# Immunogenicity and safety of a live-attenuated SARS-CoV-2 vaccine candidate based on multiple attenuation mechanisms

**Mie Suzuki Okutani[1,2†], Shinya Okamura[1,2†], Tang Gis[1], Hitomi Sasaki[1], Suni Lee[1], Akiho Kashiwabara[1,2], Simon Goto[1], Mai Matsumoto[1], Mayuko Yamawaki[1], Toshiaki Miyazaki[1], Tatsuya Nakagawa[3], Masahito Ikawa[3,4,5], Wataru Kamitani[6], Shiro Takekawa[1], Koichi Yamanishi[1], Hirotaka Ebina[1,2,4,5,7]\***

[1]The Research Foundation for Microbial Diseases of Osaka University, Suita, Japan; [2]Virus Vaccine Group, BIKEN Innovative Vaccine Research Alliance Laboratories, Institute for Open and Transdisciplinary Research Initiatives, Osaka University, Suita, Japan; [3]Department of Experimental Genome Research, Research Institute for Microbial Diseases, Osaka University, Suita, Japan; [4]Center for Advanced Modalities and DDS (CAMaD), Osaka University, Suita, Japan; [5]Center for Infectious Disease Education and Research (CiDER), Osaka University, Suita, Japan; [6]Department of Infectious Diseases and Host Defense, Gunma University Graduate School of Medicine, Maebashi, Japan; [7]Virus Vaccine Group, BIKEN Innovative Vaccine Research Alliance Laboratories, Research institute for Microbial Diseases, Osaka University, Suita, Japan

**\*For correspondence:**
hebina@biken.osaka-u.ac.jp

†These authors contributed equally to this work

## eLife Assessment

This is a **valuable** study on the efficacy of a live-attenuated vaccine that was tested in different animal models and the evidence is **convincing**. The study has been strengthened after revisions.

**Abstract** mRNA vaccines against SARS-CoV-2 were rapidly developed and were effective during the pandemic. However, some limitations remain to be resolved, such as the short-lived induced immune response and certain adverse effects. Therefore, there is an urgent need to develop new vaccines that address these issues. While live-attenuated vaccines are a highly effective modality, they pose a risk of adverse effects, including virulence reversion. In the current study, we constructed a live-attenuated vaccine candidate, BK2102, combining naturally occurring virulence-attenuating mutations in the *NSP14*, *NSP1*, spike, and *ORF7-8* coding regions. Intranasal inoculation with BK2102 induced humoral and cellular immune responses in Syrian hamsters without apparent tissue damage in the lungs, leading to protection against a SARS-CoV-2 D614G and an Omicron BA.5 strains. The neutralizing antibodies induced by BK2102 persisted for up to 364 days, which indicated that they confer long-term protection against infection. Furthermore, we confirmed the safety of BK2102 using transgenic (Tg) mice expressing human ACE2 (hACE2) that are highly susceptible to SARS-CoV-2. BK2102 did not kill the Tg mice, even when virus was administered at a dose of $10^6$ plaque-forming units (PFUs), while $10^2$ PFU of the D614G strain or an attenuated strain lacking the furin cleavage site of the spike was sufficient to kill mice. These results suggest that BK2102 is a promising live-vaccine candidate strain that confers long-term protection without significant virulence.

## Introduction

mRNA- and adenovirus vector-based vaccines have been successfully developed and used in clinical practice against SARS-CoV-2, the pathogen that caused the COVID-19 pandemic. These vaccines were highly effective, inducing robust humoral and cellular immunity (*Baden et al., 2021*; *Polack et al., 2020*). Nevertheless, certain drawbacks of SARS-CoV-2 vaccines remain to be addressed, including adverse effects, such as thrombosis, fever, and fatigue (*Yasmin et al., 2023*). Further, several boosters of the mRNA vaccines have been required to reactivate the immune response and increase efficacy against variants. New and improved vaccine modalities are therefore required for SARS-CoV-2 infection.

In general, live-attenuated vaccines are among the most effective vaccine modalities as they induce humoral and cellular immunity both systemically and locally (e.g., within mucosal surfaces), conferring protection against various infectious diseases (*Hoft et al., 2017*). For example, the live-attenuated poliovirus vaccine developed in 1962 was highly effective in reducing the spread of the disease (*Sabin, 1985*). Nevertheless, this modality has certain disadvantages, the most concerning being the risk of reversion to a virulent state as a result of mutations generated during viral replication in vivo. In fact, vaccine-derived paralytic polio was reported 38 years after the vaccine had been introduced, representing a threat to uninfected populations (*Macklin et al., 2020*). Advances in our basic knowledge of viruses have enabled the design of attenuated strains, such as that in the new type 2 oral polio vaccine (nOPV2), which is associated with a reduced risk of reversion due to genome modification (*Yeh et al., 2020*; *Yeh et al., 2023*). nOPV2 was recently approved by the World Health Organization, indicating that live-attenuated vaccines that overcome the risk of reversion are still in demand.

Various mechanisms leading to reduced pathogenicity of SARS-CoV-2 have been reported. A cold-adapted SARS-CoV-2 strain was isolated through passaging at low temperatures, eventually showing an attenuated phenotype (*Seo and Jang, 2020*). We also reported a temperature-sensitive (TS) strain with low pathogenicity and sufficient immunogenicity in vivo, achieved through the introduction of multiple TS-related mutations (*Yoshida et al., 2022*). In addition to temperature sensitivity, several naturally occurring mutants exhibit attenuated phenotypes. Passaging SARS-CoV-2 in Vero cells facilitates isolation of strains with deletions in the furin cleavage site (FCS) (PRRAR) at the S1/S2 junction within the spike protein, resulting in low proliferation in vivo (*Davidson et al., 2020*; *Peacock et al., 2021*). Furthermore, a mutant that lacks four amino acids (PRRA) within the FCS did not induce weight loss in hamsters during challenge experiments (*Hoffmann et al., 2020a*; *Hoffmann et al., 2020b*; *Johnson et al., 2021*; *Lau et al., 2020*). Reportedly, S1/S2 cleavage is essential for TMPRSS2-mediated entry into host cells, and the deletion in the FCS is thought to reduce viral growth in lung cells, which express more TMPRSS2 than the cells of the upper respiratory tract (*Hoffmann et al., 2020a*; *Hoffmann et al., 2020b*; *Johnson et al., 2021*; *Lau et al., 2020*). Furthermore, partial loss of *NSP1*, which correlates with a lower viral load and less severe symptoms of infection in SARS-CoV-2-infected patients, or of *ORF8*, which has been reported as associated with milder infection in humans, were also associated with attenuated phenotypes (*Lin et al., 2021*; *Ueno et al., 2024*; *Young et al., 2020*). Recovery of lost segments of viral genomes carrying deletions is more difficult than the reversion of amino acid substitutions (*Bull, 2015*). In this study, we designed and constructed live-attenuated vaccine candidates through a combination of substitutions and deletions involved in attenuation and reduced risk of virulent reversion. We then evaluated the candidate's immunogenicity and safety in animal models, providing evidence that BK2102 is a promising vaccine candidate that confers prolonged protection.

## Results

### Construction of the live-attenuated vaccine candidate strains

Several genomic alterations are involved in the attenuation of SARS-CoV-2. In this study, we focused on deletions at three different sites within the viral genome: FCS within the spike protein, NSP1, and ORF7-8 (*Figure 1—figure supplement 1A*). While loss of the FCS inhibits virus-cell fusion mediated by TMPRSS2 in lung cells, partial deletion of NSP1 has been shown to impair viral proliferation in vitro, and the lack of ORF8 has been associated with milder symptoms and disease outcomes (*Johnson et al., 2021*; *Lin et al., 2021*; *Ueno et al., 2024*; *Young et al., 2020*; *Zinzula, 2021*). We previously obtained SARS-CoV-2 TS strains showing diverse attenuated phenotypes and revealed that NSP3

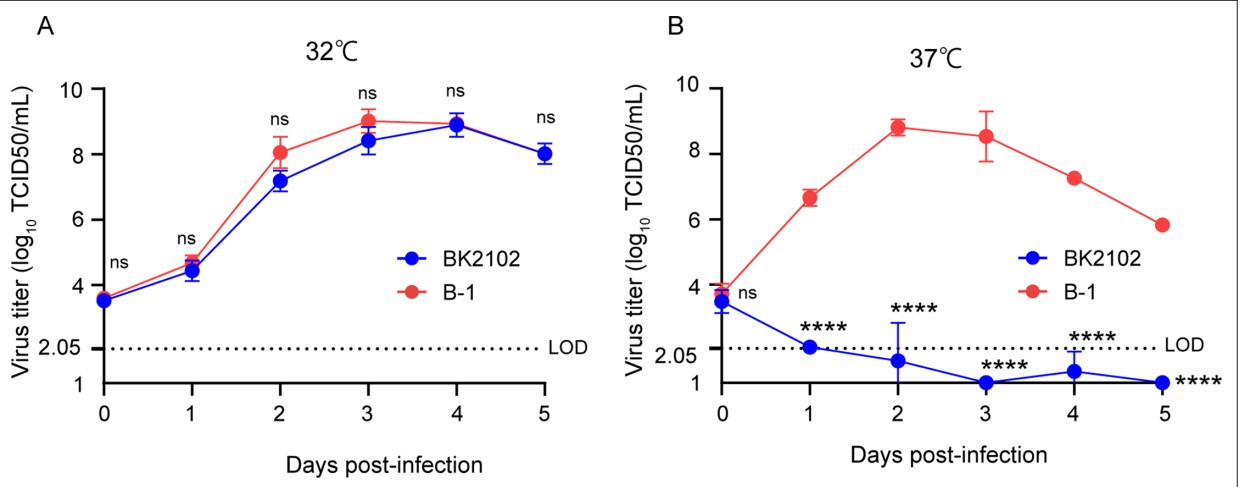

**Figure 1.** Growth dynamics of the vaccine candidate strain at different temperatures. Vero cells were infected with the wild-type parent B-1 (D614G) or the BK2102 vaccine candidate strains at a multiplicity of infection (MOI) = 0.01, and virus titers in the supernatants were determined for samples harvested every day, after incubating at 32°C (**A**) or 37°C (**B**). Infectious virus titers were determined using the TCID$_{50}$ method. Symbols indicate the average of three independent experiments, and error bars represent the SD. The limit of detection (LOD) was 2.05 log10 TCID50/mL, and for samples below the LOD, the mean value was calculated as 1 log10 TCID50/mL. The dotted line represents the assay's LOD. Days post-infection are indicated on the x-axis. For statistical analysis, two-way ANOVA with Sidak's multiple-comparison test was performed (ns, not significant; ****p<0.0001).

The online version of this article includes the following source data and figure supplement(s) for figure 1:

**Source data 1.** Related to *Figure 1A and B*.

**Figure supplement 1.** Characteristics of the vaccine candidates.

**Figure supplement 1—source data 1.** Related to *Figure 1—figure supplement 1B*.

L445F, NSP14 G248V, G416S, and A504V as well as NSP16 V67I were substitutions responsible for such phenotypes (*Yoshida et al., 2022*). Each substitution conferred some advantage for the development of an attenuated vaccine candidate with restricted proliferative capacity in deep regions of the body, such as lungs and brain. In addition, the presence of deletions is generally considered to confer a lower risk of reversion to a wild-type genotype compared to amino acid substitutions. To this end, we constructed three candidates by combining several of the above-described mutations to design a safe live-attenuated vaccine strain (*Figure 1—figure supplement 1A*). The three candidates were inoculated locally into hamsters via the nasal route to mimic a natural infection. The immunogenicity of candidates 1 and 3 was much greater than that of candidate 2 (*Figure 1—figure supplement 1B*). Candidate 1, which has three deletions in the viral genome and contains three TS-responsible substitutions in NSP14, induced neutralizing antibodies when inoculated at a dose of 10$^3$ plaque-forming unit (PFU). Candidate 2, which has TS-related substitutions in both NSP3 and NSP14 and three deletions, and Candidate 3, which has only two deletions, were speculated to be excessively attenuated or to have a higher risk of virulent reversion. Taking these observations into consideration, we selected Candidate 1 for the vaccine, hereafter referred to as BK2102. The growth dynamics of BK2102 was evaluated at 32°C and 37°C (*Figure 1*). It proliferated similarly to the wild-type B-1 strain at 32°C (*Figure 1A*), but the infectious virus titer was significantly lower compared to that of the wild-type B-1 strain 1 day post-infection at 37°C (*Figure 1B*). Therefore, BK2102 showed a severe TS phenotype but could be amplified by incubating infected cells at 32°C, which would also facilitate the manufacturing process.

## BK2102 induced humoral and cellular immune responses

To evaluate immunogenicity, BK2102 was intranasally inoculated into Syrian hamsters (10$^3$ and 10$^4$ PFU/dose). Four weeks post-inoculation, spike-specific IgG was measured in the sera by ELISA (*Figure 2A*), and the endpoint titers of the 10$^3$ PFU- and the 10$^4$ PFU-dose groups were 10$^{6.2}$ and 10$^{6.1}$, respectively. Neutralizing antibodies (*Figure 2B*) against the D614G strain were detected in 9 of the 10 hamsters in each dose group, with titers ranging between 2$^5$ and 2$^9$ (*Figure 2B*, left). Cross-reactivity of the neutralizing antibodies against the delta variant was also detected in 9 of 10 hamsters

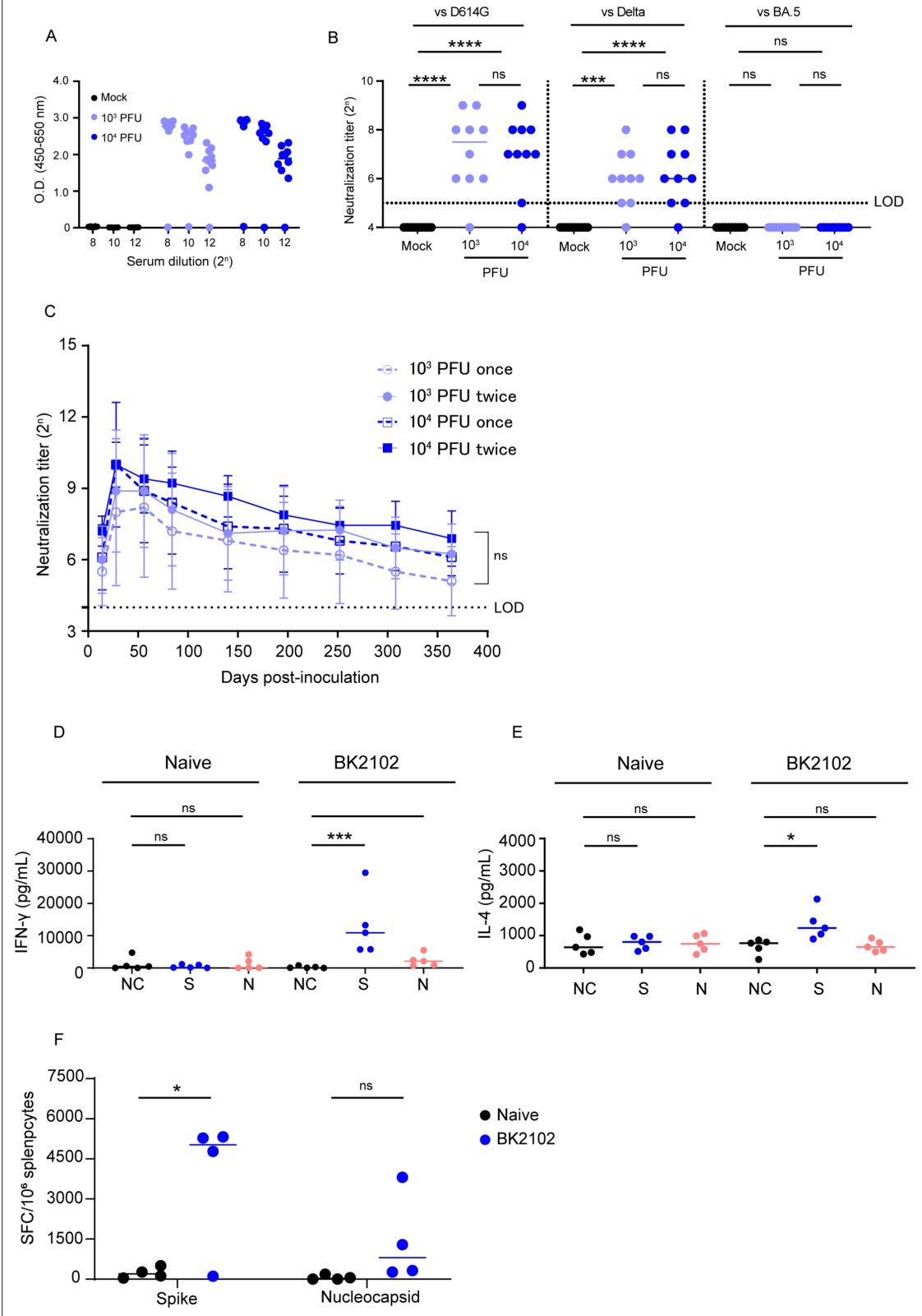

**Figure 2.** Immunogenicity of the vaccine candidate in hamsters. (**A**) Hamsters were inoculated with $1 \times 10^3$ or $1 \times 10^4$ plaque-forming unit (PFU) of BK2102 intranasally, and the serum was collected 4 weeks after inoculation. Spike-specific IgG in the sera of BK2102-inoculated hamsters and mock-treated hamsters was detected by ELISA. Symbols depict data of individual hamsters (n=10), and bars correspond to the median value. The limit of dilution is indicated in the x-axis. (**B**) Neutralizing antibodies in the sera were induced in BK2102-inoculated hamsters. Neutralizing antibodies in the

*Figure 2 continued on next page*

*Figure 2 continued*

sera were measured at day 28 post-inoculation using the following authentic SARS-CoV-2 strains: wild-type D614G (left), Delta (middle), and BA.5 (right). Symbols represent titers of individual animals (n=10), and the bars indicate the median. The limit of detection (LOD) was $2^5$, and for samples below the LOD, the mean value was set to $2^4$. The dotted line represents the assay's LOD. For statistical analysis, one-way ANOVA with Tukey's multiple-comparison test was performed (ns, not significant; ***p<0.001; ****p<0.0001). (**C**) Neutralizing antibodies persist in hamsters for at least 364 days. The neutralizing antibody titer against the authentic D614G wild-type strain was measured periodically in the sera of hamsters inoculated with BK2102 (once or twice at 4-week intervals with $1 \times 10^3$ or $1 \times 10^4$ PFU) for about a year. Symbols represent the mean of 9–10 animals, and error bars represent the SD. The LOD was $2^4$, and for samples below the LOD, the mean value was set to $2^3$. The dotted line represents the assay's LOD. For statistical analysis, two-way ANOVA with Tukey's multiple-comparison test was performed (ns, not significant). (**D, E**) Evaluation of the cellular immune response in BK2102-inoculated hamsters. Splenocytes were collected 1 week post-inoculation with $1 \times 10^4$ PFU of BK2102 and were stimulated in vitro with spike or nucleocapsid peptide pools. IFN-γ (**D**) and IL-4 (**E**) in the supernatants were measured with commercially available ELISA kits (MABTECH AB and FineTest, respectively). Symbols depict data of individual hamsters (n=5), and bars indicate the median. For statistical analysis, one-way ANOVA with Tukey's multiple-comparison test was performed (ns, not significant; *p<0.05; ***p<0.001). This experiment was conducted twice to ensure reproducibility. (**F**) Evaluation of IFN-γ-secreting cells. Four hamsters were inoculated with $1 \times 10^4$ PFU of BK2102 once and the splenocytes were collected a week later. Splenocytes were stimulated in vitro with spike or nucleocapsid peptide pools for 24 hr. IFN-γ-secreting splenocytes were quantified by ELISPOT. Symbols depict data of individual hamsters (n=4), and bars indicate the median. For statistical analysis, two-way ANOVA with Sidak's multiple-comparison test was performed (ns, not significant; *p<0.05).

The online version of this article includes the following source data and figure supplement(s) for figure 2:

**Source data 1.** Related to *Figure 2A*.

**Source data 2.** Related to *Figure 2B*.

**Source data 3.** Related to *Figure 2C* and *Figure 2—figure supplement 3*.

**Source data 4.** Related to *Figure 2D and E*.

**Source data 5.** Related to *Figure 2F*.

**Figure supplement 1.** Comparison of immunogenicity of BK2102 with other vaccine modalities.

**Figure supplement 1—source data 1.** Related to *Figure 2—figure supplement 1A*.

**Figure supplement 1—source data 2.** Related to *Figure 2—figure supplement 1B*.

**Figure supplement 1—source data 3.** Related to *Figure 2—figure supplement 1C*.

**Figure supplement 1—source data 4.** Related to *Figure 2—figure supplement 1D*.

**Figure supplement 2.** Immunogenicity of BK2102 in monkeys.

**Figure supplement 2—source data 1.** Related to *Figure 2—figure supplement 2A*.

**Figure supplement 2—source data 2.** Related to *Figure 2—figure supplement 2B*.

**Figure supplement 3.** Persistence of the neutralizing antibodies induced by BK2102 in each group.

**Figure supplement 4.** BK2102 induces protective immunity against SARS-CoV-2 gamma strain.

**Figure supplement 4—source data 1.** Related to *Figure 2—figure supplement 4A*.

(titer range: $2^5$–$2^8$) (**Figure 2B**, middle) and 8 of 9 hamsters against gamma strain (**Figure 2—figure supplement 4A**), but that against the BA.5 variant was below the limit of detection in all hamsters (**Figure 2B**, right). Furthermore, we performed BK2102 immunization of cynomolgus monkeys at a dose of $10^7$ PFU, and the serum neutralizing titer against the D614G strain was detected in two of the four monkeys (titer range: $2^4$–$2^8$) (**Figure 2—figure supplement 2A**). Although a single dose did not raise neutralizing antibody titers in two of the four monkeys, three doses given at a 2-week interval induced neutralizing antibodies in all six monkeys (**Figure 2—figure supplement 2B**). The safety of BK2102 was also evaluated in these six monkeys, and no toxic effects were observed in any of the parameters assessed, including tissue damage, respiratory rate, functional observational battery (FOB), hematology, or fever (data not shown).

A short-lived immune response has been reported for current mRNA vaccines against SARS-CoV-2. For example, a reduction in neutralizing antibodies was observed in humans after 6 months (*Zhang et al., 2022*). In hamsters, these were undetectable after 250 days (*Machado et al., 2023*). Therefore, we evaluated the persistence of the immune response induced by BK2102 using a hamster model. We measured neutralizing antibody titers for up to 364 days after inoculation with BK2102. The titer peaks were observed 28 days after the first inoculation and slightly decreased, but were maintained until 364 days post-inoculation with a dose of $10^3$ or $10^4$ PFU. For example, the neutralizing antibody titer in the sera of hamsters inoculated with $10^3$ PFU was $2^8$ at 28 days and $2^5$ at 364 days post-inoculation.

Also, 2 of the 10 hamsters inoculated at a dose of $10^3$ PFU showed neutralizing antibody titers below the detection limit from the beginning, and those of another hamster in the same dose group began to decrease gradually from day 224 and fell below the detection limit on day 364 (*Figure 2—figure supplement 3*). However, the neutralizing antibody titers in hamsters inoculated with $10^4$ PFU did not exhibit such a decrease during the evaluation period. Remarkably, a single dose of BK2102 was sufficient to induce long-lasting immunity, and there was no need for booster immunization 28 days after the first inoculation (*Figure 2C*).

Furthermore, we evaluated cellular immune responses following inoculation with BK2102. Antigen-specific IFN-γ and IL-4 production in spleen cells from inoculated hamsters was measured via ELISA after in vitro re-stimulation with spike or nucleocapsid peptides. As shown in *Figure 2D*, spike peptide-specific IFN-γ production significantly increased in the splenocytes of BK2102-inoculated hamsters, as did the nucleocapsid peptide-specific IFN-γ production, although in this case it did not reach statistical significance. IL-4 production was significantly increased by spike-peptide stimulation (*Figure 2E*). IFN-γ-producing cells were also detected by enzyme-linked immunosorbent spot (ELISPOT) assays (*Figure 2F*). In correlation with the ELISA results, significant induction of spike peptide-specific IFN-γ-producing cells were detected in the splenocytes of BK2102-inoculated hamsters and the nucleocapsid peptide-specific IFN-γ-producing cells were also increased.

Moreover, a 10-μg-dose of a conventional mRNA vaccine prepared in-house, expressing spike protein of D614G strain of SARS-CoV-2, was intramuscularly injected into hamsters and compared to BK2102. Neutralizing antibody titers against the D614G strain showed no significant difference between the BK2102-inoculated group and the mRNA vaccine group (*Figure 2—figure supplement 1A*). Notably, under conditions that induced equal serum neutralizing antibody titers in hamsters, higher levels of spike-specific IgA in nasal wash samples were induced by BK2102 than by the conventional mRNA (*Figure 2—figure supplement 1B*). In addition, we qualitatively analyzed the spike-specific IgG subclasses (IgG1 and IgG2/3) in hamsters to evaluate the nature of the immune response induced by BK2102 (*Figure 2—figure supplement 1C and D*). When we inoculated BK2102 and our mRNA vaccine, total IgG antibodies were detected in both groups. IgG2/3 antibodies were detected in the sera of BK2102-inoculated hamsters, but we could not detect IgG1. On the other hand, mRNA-vaccinated hamsters showed both IgG subclasses (*Figure 2—figure supplement 1C*). As other studies have demonstrated that aluminum adjuvant preferentially induces a Th2 response (*Marrack et al., 2009*), we also administered recombinant spike protein with alum adjuvant as a control. The result was the same since BK2102-inoculated hamsters showed only production of IgG2/3 (*Figure 2—figure supplement 1D*). IL-4 production and the presence of IgG1 reflects a Th2 response, while IFN-γ production and IgG2/3 are indicative of a Th1 response in hamsters (*Kushawaha et al., 2011*; *Ploquin et al., 2013*). Our results therefore suggest that BK2102 mainly induced a Th1 immune response in hamsters.

## BK2102 induced protective immunity against SARS-CoV-2 infection

Next, we performed challenge experiments with the SARS-CoV-2 D614G strain, BA.5 or gamma variants in order to evaluate whether the immune responses induced by BK2102 would protect against infection. All hamsters inoculated with BK2102 did not lose weight, whereas the naïve hamsters lost approximately 10% of their total body weight on day 4 or 6 post-challenge with the D614G or gamma strains (*Figure 3* and *Figure 3—figure supplement 1*, respectively). When challenged with the BA.5 variant, all hamsters pre-inoculated with a dose of $10^4$ PFU of BK2102 and three of five hamsters pre-inoculated with a dose of $10^3$ PFU did not lose weight. However, the rest of the animals in the $10^3$ PFU dose group lost 5% of their total body weight, similarly to the naïve group (*Figure 3B*).

In addition to body weight change, we determined infectious virus titers in lung homogenates and nasal wash specimens 4 days-post infection with the D614G strain or the BA.5 variant. The number of infectious viruses was significantly lower in hamsters inoculated with BK2102 than in naïve hamsters after challenge with the D614G strain (*Figure 3C and D*). One hamster in the $10^3$ PFU dose group showed detectable levels of infectious virus following D614G challenge (*Figure 3C*). This result was consistent with the undetectable levels of neutralizing antibodies in this animal (*Figure 2B*). The virus titer in the lung and nasal wash of one animal in the $10^3$ PFU dose group was 6.4 $\log_{10}$ PFU/g and 3.8 $\log_{10}$ PFU/mL, respectively, and the mean virus titers in the naïve group was 6.1 $\log_{10}$ PFU/g and 3.7 $\log_{10}$ PFU/mL, respectively. Although the cross-reactivity of neutralizing antibodies against the

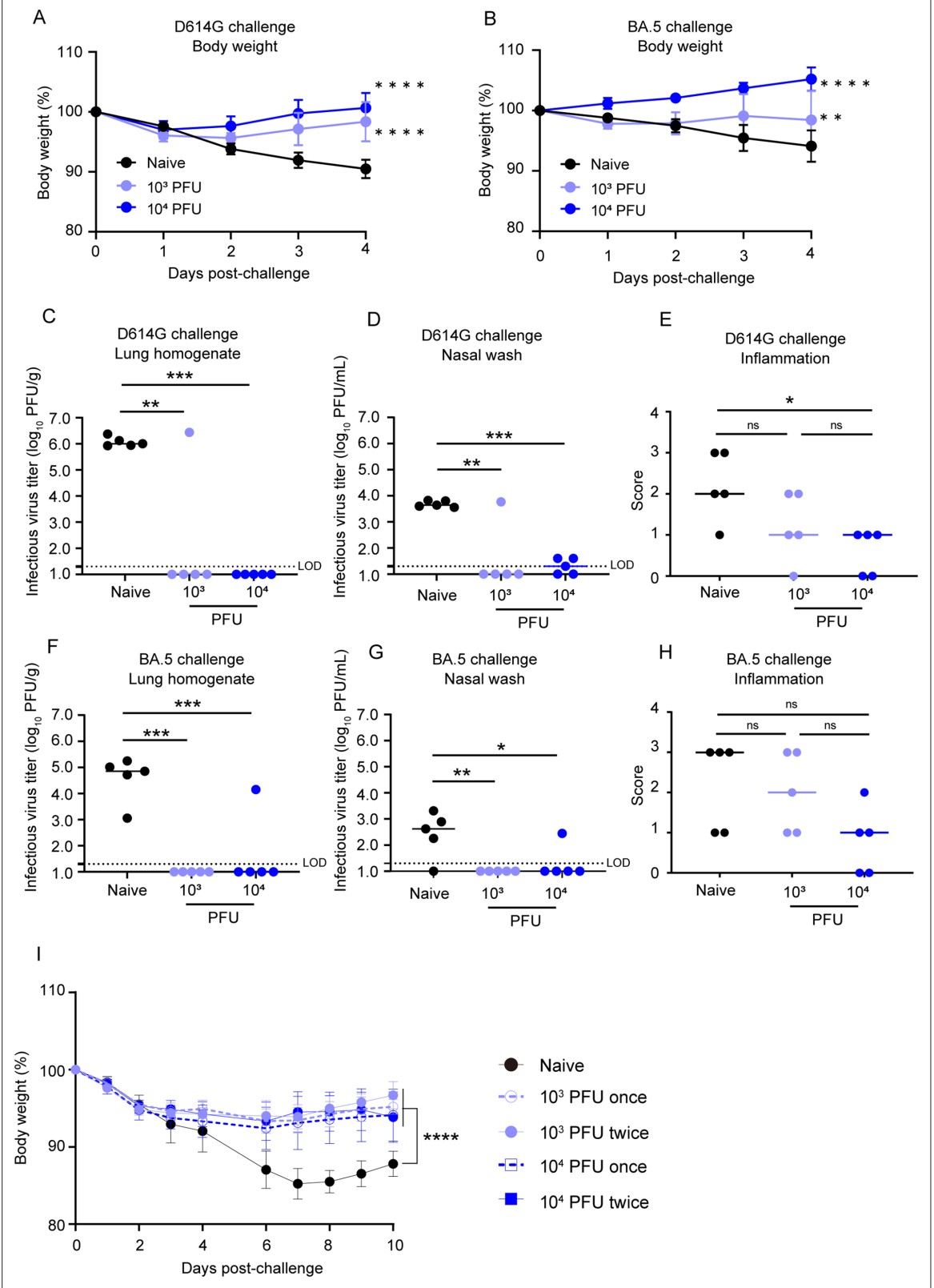

**Figure 3.** BK2102 induces protective immunity. (**A, B**) BK2102 protects hamsters against homologous and heterologous virus challenges. Hamsters that received a full vaccination protocol with the indicated doses of BK2102 were challenged with $3 \times 10^5$ plaque-forming unit (PFU) of wild-type D614G (**A**) or BA.5 (**B**) strains, and their body weight was monitored for 4 days. Body weight is expressed as a percentage of the initial weight. The symbols represent the average weight of the hamsters (n=5), and error bars indicate the mean SD. Two-way ANOVA with Tukey's multiple-comparison test

*Figure 3 continued on next page*

*Figure 3 continued*

was performed for statistical analysis (**p<0.01; ****p<0.0001). (**C, D, F, G**) The infectious virus titer in the lungs and nasal wash specimens taken on day 4 post-challenge was measured via a plaque assay for the wild-type D614G strain (**C, D**) and for the BA.5 strain (**F, G**). Symbols represent titers of individual animals (n=5), and the bars indicate the median. The limit of detection (LOD) was 1.3 $\log_{10}$ PFU/g or PFU/mL, and for samples below the LOD, the mean value was calculated as 1.0 $\log_{10}$ PFU/g or PFU/mL. The dotted line represents the assay's LOD. One-way ANOVA with Dunnett's multiple-comparison test was performed for statistical analysis (ns, not significant; *p<0.05; **p<0.01; ***p<0.001). (**E, H**) Lung inflammation scores were determined via H&E staining of D614G- (**E**) and BA.5-challenged (**H**) hamsters. The percentage of the disrupted area in the entire visual field was classified as 0: not remarkable (<10%); 1: minimal (10–50%); and 2: mild (50–70%). Symbols depict data of individual animals (n=5), and the bars indicate the median. One-way ANOVA with Tukey's multiple-comparison test was performed for statistical analysis (ns, not significant; *p<0.05). (**I**) Weight changes after the challenge assay 1 year post-inoculation with BK2102. Hamsters inoculated with BK2102 were challenged with the wild-type D614G strain at $3 \times 10^5$ PFU on 420 days. Nine-month-old elder hamsters were used as the naïve group. The symbols represent the average weight of the hamsters (n=4 or 5), and error bars indicate the mean SD. Two-way ANOVA with Tukey's multiple-comparison test was performed for statistical analysis (****p<0.0001).

The online version of this article includes the following source data and figure supplement(s) for figure 3:

**Source data 1.** Related to *Figure 3A and B*.

**Source data 2.** Related to *Figure 3C, D, F and G*.

**Source data 3.** Related to *Figure 3E and H*.

**Source data 4.** Related to *Figure 3I*.

**Figure supplement 1.** BK2102 induces protective immunity against SARS-CoV-2 gamma strain.

**Figure supplement 1—source data 1.** Related to *Figure 3—figure supplement 1B*.

**Figure supplement 2.** Evaluation of BK2102 onward transmission in hamsters.

**Figure supplement 2—source data 1.** Related to *Figure 3—figure supplement 2B*.

BA.5 variant was below the limit of detection in all hamsters (*Figure 2B*, right), no infectious virus was detected following challenge with the BA.5 variant in most of the vaccinated animals (*Figure 3F and G*). The virus titer in the lung and nasal wash of one animal in $10^4$ PFU dose group was 4.2 $\log_{10}$ PFU/g and 2.5 $\log_{10}$ PFU/mL, respectively, and the mean virus titers in the naïve group where virus was detected were 4.6 $\log_{10}$ PFU/g and 2.8 $\log_{10}$ PFU/mL, respectively. Lung tissue damage after the viral challenge was also evaluated in the hamsters. The inflammation score of hamsters inoculated with BK2102 was lower than that of naïve hamsters, regardless of the strain/variant used for the challenge (*Figure 3E and H*, respectively). These results suggest that the immune response induced by BK2102 confers protection against infection that is not limited to the SARS-CoV-2 D614G strain, but also includes the BA.5 variant.

Furthermore, to evaluate whether the protection conferred after a full vaccination protocol would persist over time, hamsters with confirmed persistent immunity at day 364 post-inoculation (*Figure 2C*) were challenged at day 420. In elderly naïve hamsters, a body weight loss of approximately 15% was observed 7 days after infection with the D614G strain, whereas the BK2102-inoculated hamsters showed a lower weight decrease, with significant differences noted at this time point (*Figure 3I*). Therefore, BK2102 induced a prolonged humoral immune response, which contributed to the protection against viral infection in hamsters.

We then evaluated whether BK2102 could inhibit onward transmission, as a previous report of a live-attenuated vaccine generated through codon-pair deoptimization (sCPD9) had suggested that an effective immune response within the nasal cavity would likely prevent it (*Nouailles et al., 2023*). The naïve group, the spike protein-inoculated (intra-muscularly) group, and the BK2102 intranasal inoculation groups were challenged with the SARS-CoV-2 D614G strain and co-housed with another group of naïve hamsters 1 day later (*Figure 3—figure supplement 2A*). Naïve animals co-housed with hamsters in the naïve or intramuscularly spike-alum vaccinated groups showed slight weight loss. However, no weight loss was observed in hamsters co-housed with the hamsters that had been intranasally inoculated with BK2102 (*Figure 3—figure supplement 2B*). Therefore, intranasal inoculation of the BK2102 live-attenuated vaccine effectively prevented onward transmission, in line with a previous report (*Nouailles et al., 2023*).

## BK2102 caused localized tissue damage and conferred a low risk of transmission

Next, we evaluated the safety of BK2102 by assessing the tissue damage during acute infection. The lungs and whole heads of hamsters at day 3 post-infection were extracted and fixed with formalin (*Figure 4A*). We evaluated inflammation and detected viral antigens in the lungs and multiple-depth sections of the nasal cavity (*Figure 4B and C* and *Figure 4—figure supplement 1*). The D614G strain caused broad inflammation within the nasal cavity (from level 1 to 3) and lungs. Viral antigens were detected in the same areas, consistent with our previous report (*Yoshida et al., 2022*). However, in the BK2102-infected hamsters, viral antigens and weak-to-mild tissue damage were observed only in the anterior area of the nasal cavity, whereas no tissue damage or viral antigens were detected in the posterior area or lungs.

The replication of BK2102 at the tip of the nasal cavity may facilitate transmission because infectious viruses are shed into the nasal fluid. In addition, virulent reversion may occur during replication in vivo. Therefore, we evaluated the risk of transmission and reversion to virulence by passaging in vivo using hamsters (*Figure 4D*). SARS-CoV-2 A50-18 is a previously isolated TS strain, in which substitutions within the NSP14 protein alone account for the TS phenotype, without the need for deletions, such as those in NSP1, spike, or other accessory proteins (*Yoshida et al., 2022*). The viral genome was detected in all nasal wash specimens from A50-18 strain-infected hamsters during primary infection, and an increase in this amount was observed in subsequently passaged samples (*Figure 4E*), which correlated with progressive weight loss (*Figure 4—figure supplement 2*). When we confirmed the sequence of the viruses detected in samples, we observed that the TS-responsible substitutions in NSP14 had reverted to the wild-type sequence (*Table 1*). The viral genome was detected in the nasal wash specimens from BK2102-infected hamsters during primary infection, but we could not detect it in the samples from subsequent hamsters, except for in one case in p-1 (*Figure 4E*). In this individual animal, the viral genome was not detected in the samples from later passages. No weight loss was observed in any of the primary or subsequently infected hamsters, and we did not detect changes to the wild-type sequence (*Table 1*, *Figure 4—figure supplement 2*). Overall, our results suggest that BK2102 is a safe live-attenuated vaccine candidate with a low risk of virulent reversion.

## BK2102 showed a favorable safety profile in Tg mice

hACE2 Tg mice are also used as animal models of SARS-CoV-2 infection (*Asaka et al., 2021*; *Bao et al., 2020*). We established a mouse line expressing hACE2 driven by the CAG promoter, and these mice were used to evaluate the safety of BK2102 live-attenuated vaccine candidate. hACE2 expression was detected not only in the respiratory tract, but also in various tissues such as the central nervous system, heart, skeletal muscle, digestive system (except the small intestine), spleen, and testis (*Figure 5—figure supplement 1A*). We evaluated the survival rates and body weight of Tg mice infected with various SARS-CoV-2 strains. All of the mice died after weight loss by infection with the D614G strain and even with the FCS deleted B-1 (B-1 ΔFCS) strain, previously established attenuated phenotype, within 6 days after receiving a dose of $10^2$ PFU (*Figure 5* and *Figure 5—figure supplement 1*). Meanwhile, a higher survival rate was observed in mice infected with the L50-33 and A50-18 strains, which were previously isolated TS and live-attenuated strains (*Yoshida et al., 2022*), even at a dose of $10^5$ PFU. However, one mouse in each group infected with $10^4$ and $10^5$ PFU of the L50-33 strain died 10 days post-infection that is 4 days later than those in the D614G strain-infected group. This time lag before death was expected as the virus could have reverted during replication in vivo, being able to grow in deeper regions of the body. Thus, we evaluated the presence of infectious virus in the lungs and brains of mice that died following infection with the D614G, B-1 ΔFCS, and L50-33 strains. Infectious virus titers in the lungs were approximately 2.40–5.64 $\log_{10}$ PFU/g, while those in the brains were higher, at approximately 5.75–8.69 $\log_{10}$ PFU/g (*Table 2*). We noticed that mice exhibiting head nodding, intense running, jumping and repeated falling died despite a generally mild inflammation in the lungs at necropsy (data not shown). These results suggest that Tg mice were killed due to replication of SARS-CoV-2 in the brain rather than in the lungs. Sanger sequencing analysis of viruses in the lungs and brains of mice that died following infection with the L50-33 strain revealed that the 445F substitution in NSP3, responsible for the TS phenotype of this strain, had reverted to the wild-type amino acid leucine (TTT → TTG), which is the same in the D614G and B-1 ΔFCS strains (CTT, *Table 2*). These results suggest that the mice died due to viral proliferation in the brain, where a small

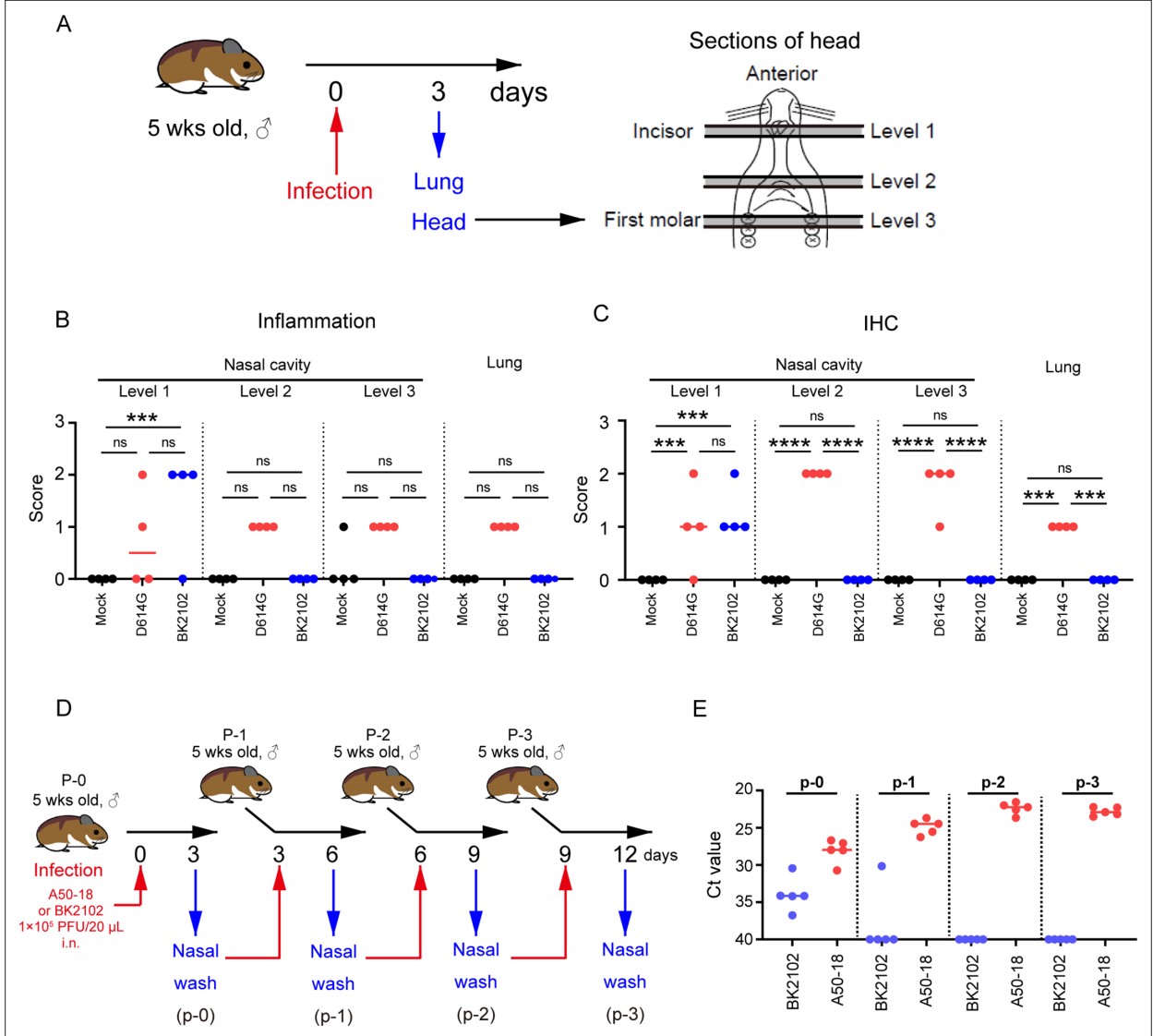

**Figure 4.** BK2102 caused localized tissue damage and posed as low risk of transmission. (**A**) Scheme for the evaluation of tissue damage in acute infection with BK2102 in a hamster model. The wild-type D614G strain was used as a positive control. (**B**) Inflammation score of nasal cavity sections and lungs determined via H&E. The percentage of the disrupted area in the entire visual field was classified as 0: not remarkable (<10%); 1: minimal (10–50%); 2: mild (50–70%), respectively. The symbols depict data of individual hamsters (n=4), and the bars indicate the median. One-way ANOVA with Tukey's multiple-comparison test was performed for statistical analysis (ns, not significant; ***p<0.001). (**C**) SARS-CoV-2 spike protein staining in the nasal cavity sections and lungs determined via immunohistochemistry using a SARS-CoV-2 spike RBD-specific antibody. The proportion of positive cells in the entire visual field was classified as 0: not remarkable (<10%); 1: minimal (10–50%); and 2: mild (50–70%), respectively. The symbols depict data of individual hamsters (n=4), and the bars indicate the median. One-way ANOVA with Tukey's multiple-comparison test was performed for statistical analysis (ns, not significant; ***p<0.001; ****p<0.0001). (**D**) Scheme for the evaluation of BK2102 transmission via in vivo passage in hamsters. The TS-strain A50-18 was used as a positive control. (**E**) Ct values obtained for the RT-PCR performed using RNA extracted from the nasal wash specimens. This experiment was conducted three times to ensure reproducibility. The symbols depict data of individual hamsters (n=5), and the bars indicate the median.

The online version of this article includes the following source data and figure supplement(s) for figure 4:

**Source data 1.** Related to *Figure 4B and C*.

**Source data 2.** Related to *Figure 4E*.

**Figure supplement 1.** Evaluation of the tissue damage induced by BK2102.

**Figure supplement 2.** BK2102 showed a low risk of transmission.

**Figure supplement 2—source data 1.** Related to *Figure 4—figure supplement 2*.

**Table 1.** Genetic variations of viruses passaged in vivo.

|  | Mutations | | | | | |
|---|---|---|---|---|---|---|
|  | ΔG359 - A382 (*NSP1*) | G18782T (*NSP14*) | G19285A (*NSP14*) | C19550T (*NSP14*) | ΔA23598-G23624 (*Spike* FCS) | ΔC27549-T28251 (*ORF7a-8*) |
| A50-18 | N/A | G | G/A | T | N/A | N/A |
|  | N/A | G/T | G/A | T | N/A | N/A |
|  | N/A | G/T | G/A | T | N/A | N/A |
|  | N/A | G/T | G | T | N/A | N/A |
|  | N/A | G/T | G/A | T | N/A | N/A |
| BK2102 | O* | T | A | T | O* | O* |
|  | O* | T | A | T | O* | O* |
|  | O* | T | A | T | O* | O* |
|  | O* | T | A | T | O* | O* |
|  | O* | T | A | T | O* | O* |

N/A, not applicable.

*Same sequence as the inoculated virus.

The online version of this article includes the following source data for table 1:

**Source data 1.** AB1 files of sequence data for *Table 1*.

**Source data 2.** Agarose gel electrophoresis pattern of PCR products corresponding to the amplification of the ORF7a-8.

virus population lost its temperature sensitivity, becoming virulent. This mouse is a highly susceptible model of SARS-CoV-2 virus infection able to detect a few TS revertant viruses. In contrast to L50-33 with the NSP3-based TS phenotype, A50-18, harboring three TS-responsible substitutions in NSP14, did not kill any mice. Moreover, in the case of BK2102, no mice died or lost weight following infection, even at a dose of $10^6$ PFU (*Figure 5* and *Figure 5—figure supplement 1*, respectively). Therefore, BK2102 is considered to have a low risk of virulent reversion, thus representing a suitable candidate for a safe live-attenuated vaccine.

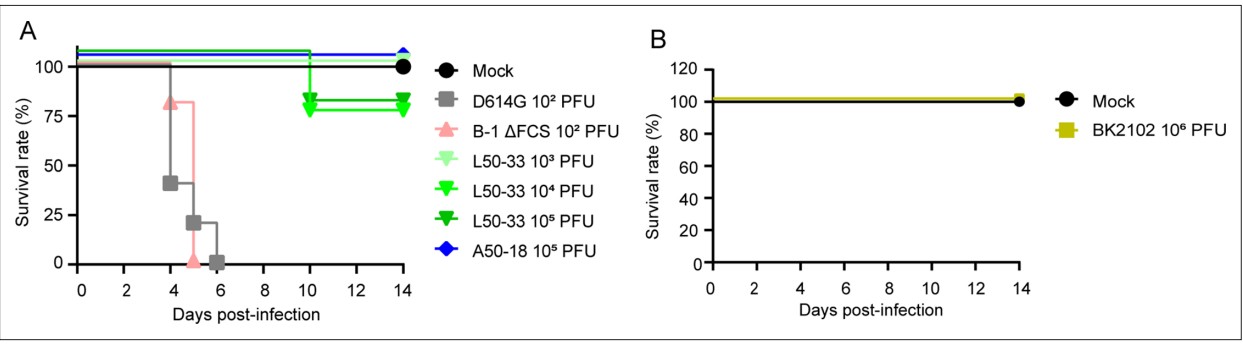

**Figure 5.** BK2102 showed a highly safe phenotype in Tg mice. (**A, B**) Survival rate of Tg mice infected with the wild-type D614G, B-1 ΔFCS, L50-33, and A50-18 TS strains (**A**) and BK2102 (**B**). The symbols represent the survival rate of the hamsters (n=4 or 5).

The online version of this article includes the following source data and figure supplement(s) for figure 5:

**Source data 1.** Related to *Figure 5* and *Figure 5—figure supplement 1*.

**Source data 2.** Related to *Figure 5* and *Figure 5—figure supplement 1*.

**Figure supplement 1.** Expression of hACE2 and body weight of Tg mice after infection.

**Figure supplement 1—source data 1.** Original western blots for *Figure 5—figure supplement 1*.

**Figure supplement 1—source data 2.** Image raw data of western blots for *Figure 5—figure supplement 1*.

**Table 2.** NSP3 genetic variations in viruses recovered from infected Tg mice.

| | n | Virus titer (Log10 PFU/g) | | TS substitution |
| | | Brain | Lung | NSP3 |
| --- | --- | --- | --- | --- |
| D614G (10$^2$ PFU infected) | 5 | 6.94 ± 0.44* | 3.67 ± 0.26* | N/A |
| B-1 ΔFCS (10$^2$ PFU infected) | 5 | 8.69 ± 0.33* | 5.64 ± 0.62* | N/A |
| L50-33 (10$^4$ PFU infected) | 1 | 5.75 | <2.40 | F445L (TTT → TTG) |
| L50-33 (10$^5$ PFU infected) | 1 | 7.24 | 2.40 | F445L (TTT → TTG) |

N/A, not applicable.
*Average ± SD.

The online version of this article includes the following source data for table 2:

**Source data 1.** Virus titer for *Table 2*.

**Source data 2.** AB1 files of sequence data for *Table 2*.

## Discussion

In this study, we evaluated candidates for a new live-attenuated SARS-CoV-2 vaccine. The candidate harboring more mutations exhibited lower immunogenicity than those with fewer mutations (*Figure 1—figure supplement 1A*), presumably because excessive mutations hinder viral replication in the body, resulting in weaker immune responses. Deletions within viral genes are less prone to reversion to the wild-type genotype, and TS-associated substitutions restrict viral dissemination to deep or warmer regions of the body, including the brain, which makes them an attractive backbone for developing safe live-attenuated vaccines. Our BK2102 vaccine candidate was designed with three deletions in addition to TS-associated substitutions.

We previously reported that TS strains replicate only in the anterior regions of the nasal cavity and do not proliferate in the posterior areas or lungs of hamsters. However, these isolated TS strains posed a risk of reversion to virulence (*Yoshida et al., 2022*). K18-hACE2 Tg mice have been shown to die after infection with several SARS-CoV-2 strains due to viral proliferation in the brain (*Natekar et al., 2022*). To better assess the safety of BK2102, we generated and used CAG-hACE2 Tg mice in addition to hamsters. The TS strains (L50-33 and A50-18) showed an attenuated phenotype in this mouse model, emphasizing the ability of TS to limit viral replication and restrict replication-permissive regions by temperature. The TS-responsible substitutions in NSP14 were more stable than those in NSP3, as L50-33 caused the death of one mouse in each of the dose groups tested, whereas A50-18 did not. Additionally, infectious viruses lacking TS mutations were detected in the central nervous system and respiratory tracts of mice that died following L50-33 infection (*Table 2*). These findings suggest that our mouse model is particularly suitable for assessing safety and evaluating TS strains' entry into the nervous system as it enables the detection of even a small population of revertant viruses. This animal model also addresses regarding SARS-CoV-2 infection via the intranasal route causing central nervous system damage (*Jha et al., 2021*; *Kumar et al., 2020*).

BK2102 did not proliferate in the brains of Tg mice even at a dose of 10$^6$ PFU, which was 10,000 times higher than the dose that killed mice infected with the D614G or B-1 ΔFCS strains (*Figure 5A and B*). Importantly, BK2102 was not detected during passaging in vivo using naïve hamsters, probably due to the multiple defective mutations that controlled replication in the host animal, preventing sufficient proliferation for transmission. In contrast, the A50-18 strain, with a genetic background similar to BK2102 but lacking the three deletion mutations, showed an attenuated phenotype during primary infection, but reverted to virulence during replication in hamsters, leading to transmission of the virulent strain to naive animals (*Figure 4E*). We hypothesized that even if TS-related substitutions are lost during primary infection, the remaining deletion mutations in BK2102 ensure low replication efficiency, preventing transmission. Indeed, mutations in the FCS coding region have been reported

to reduce proliferation (*Lau et al., 2020*; *Sasaki et al., 2021*; *Wang et al., 2021a*) and prevent transmission in vivo (*Peacock et al., 2021*).

Our vaccine candidate BK2102 induced humoral and cellular immune responses in hamsters, and animals were protected against challenge with the heterologous BA.5 variant, even though the neutralizing titer in serum was below the limit of detection (*Figure 2B*, right). The nucleocapsid proteins of many coronaviruses are highly immunogenic and are abundantly expressed in infected cells, making them effective targets for antigen-specific T cells (*Cong et al., 2020*; *Dutta et al., 2020*; *Hasanpourghadi et al., 2023*). Studies have also reported the benefit of vaccination with the nucleocapsid protein of SARS-CoV-2, showing protective immunity in animal models vaccinated with the nucleocapsid protein alone (*Primard et al., 2023*) or in combination with the spike protein (*Chiuppesi et al., 2022*; *Hasanpourghadi et al., 2023*). In our hamster study, we observed cellular immune responses against both the nucleocapsid and spike protein. BK2102 may induce a cellular immune response against various structural proteins of SARS-CoV-2, providing protection against multiple variants. We also considered the potential induction of mucosal immunity. In this study, BK2102 induced spike-specific IgA in nasal wash (*Figure 2—figure supplement 1B*). Live-attenuated influenza vaccines have been reported to be effective against a broad range of variants due to the robust humoral and cellular immune responses they elicit, even in mucosal tissues (*Thwaites et al., 2023*). BK2102 was administered intranasally, and inhibition of onward transmission was observed (*Figure 3—figure supplement 2*), similar to what was reported for the SARS-CoV-2 live-attenuated vaccine candidate sCPD9 (*Nouailles et al., 2023*). Thus, we assumed that BK2102 also induced a mucosal immune response in addition to systemic humoral and cellular immunity, contributing to protection against mutant strains and prevention of onward transmission to other animals.

The findings of this study should be interpreted in light of certain limitations. First, the limited availability of analytical reagents for hamster models restricted the detailed immunological characterization of the response. Additionally, it took time to gather preclinical data due to the space-related restrictions of BSL3 facilities, which delayed the clinical trials for BK2102 until many individuals had already acquired immunity against SARS-CoV-2. It remains to be seen whether our candidate will be optimal for human use as the immunogenicity of live-attenuated vaccines is generally influenced by pre-existing immunity. Finally, species-related differences in susceptibility must also be considered. The minimum infectious titer of SARS-CoV-2 has been reported as 10 TCID50 in hamsters and humans (*Lindeboom et al., 2024*; *Rosenke et al., 2020*), but $3.84 \times 10^4$ PFU in monkeys (*Johnston et al., 2021*). When we inoculated this candidate into monkeys, they were less susceptible than hamsters, requiring a very high titer and multiple doses to induce immunity. Therefore, the dosage for first-in-human trials should be carefully optimized. It is also important to note that live-attenuated vaccines are contraindicated in immunosuppressed individuals or those with chronic diseases, and BK2102 is not intended for these populations.

The key to developing live-attenuated vaccines lies in balancing immunogenicity and safety. Most live-attenuated vaccine candidates, such as sCPD9 and CoviLiv (which is in phase III clinical trial), have been evaluated with a primary focus on immunogenicity in animal models (*Wang et al., 2021b*). In this study, we rigorously assessed not only the immunogenicity, but also the safety of BK2102, demonstrating its superior safety profile. For example, sCPD9 and CoviLiv are attenuated by codon deoptimization or a combination of codon deoptimization and FCS deletion (*Trimpert et al., 2021*; *Wang et al., 2021b*). These strategies affect viral proliferation but not necessarily virulence. The TS-responsible substitutions in NSP14 included in BK2102 selectively restrict the infection site, reducing the likelihood of lung and brain infection and enhancing safety.

Our findings also indicated that the TS substitutions in BK2102 made it difficult to passage the virus in vivo, limiting its spread to the central nervous system. Although attenuated strains with amino acid substitutions pose a risk of reversion to virulence, combining multiple modifications related to diverse attenuated phenotypes may allow for the construction of safer live-attenuated vaccine candidates. Among these modifications, deletions are particularly useful in reducing the risk of reversion to virulence, and TS-responsible substitutions significantly limit viral replication in deep regions of the body. This strategy of combining multiple modifications could be effective for the development of live-attenuated vaccines against other viruses, such as the Japanese encephalitis and influenza viruses, which have been reported to replicate in the brain and lung (*Desai et al., 1995*; *Weinheimer et al., 2012*).

**Table 3.** SARS-CoV-2 strains.

| Reagent or resource | | Source | Identifier |
|---|---|---|---|
| SARS-CoV-2: pre-alpha type, D614G, B-1 strain | | *Yoshida et al., 2022* | NCBI: LC603286 |
| | A50-18 strain | *Yoshida et al., 2022* | NCBI: LC603287 |
| | L50-33 strain | *Yoshida et al., 2022* | NCBI: LC603289 |
| | B-1 ΔFCS strain | Current study | N/A |
| | Candidate 1 (BK2102) | Current study | N/A |
| | Candidate 2 | Current study | N/A |
| | Candidate 3 | Current study | N/A |
| SARS-CoV-2: delta variant, BK325 strain | | Research Foundation for Microbial Diseases of Osaka University | N/A |
| SARS-CoV-2: gamma variant, TY7-501 strain | | National Institute of Infectious Diseases | GISAID ID: EPI_ISL_833366 |
| SARS-CoV-2: omicron variant, TY41-702 strain | | National Institute of Infectious Diseases | GISAID ID: EPI_ISL_13241867 |

N/A, not applicable.

## Materials and methods

### Cells and viruses

Vero cells (Cat# CCL-81) were purchased from ATCC and maintained in D-MEM supplemented with 10% FBS, penicillin (100 U/mL), and streptomycin (0.1 mg/mL). Cells were confirmed for mycoplasma contamination using a e-Myco VALiD Mycoplasma PCR Detection Kit (iNtRON Biotechnology, Inc, 25239). VeroE6/TMPRSS2 cells (no. JCRB1819) were obtained from the Japanese Collection of Research Bioresources (JCRB) cell bank and cultured in D-MEM supplemented with 10% FBS, penicillin (100 U/mL), streptomycin (0.1 mg/mL), and G-418 (1 mg/mL). We previously constructed baby hamster kidney (BHK) cells constitutively expressing the human angiotensin-converting enzyme 2 (hACE2) (BHK/hACE2 cells) (*Okamura et al., 2023*), and these cells were maintained in MEM, supplemented with 10% FBS, penicillin-streptomycin, and puromycin (3 μg/mL). 293T-ACE2 cells were purchased from Abnova Corporation as COVID-19 Pseudovirus Neutralizing Antibody Assay kit (Cat# KA6152) and cultured in D-MEM supplemented with 10% FBS. The SARS-CoV-2 strains used in this study are listed in *Table 3*. The SARS-CoV-2 B-1 (D614G) strain was isolated from a clinical specimen, and TS derivative strains were obtained through random mutagenesis of this clinical isolate, as previously reported (*Yoshida et al., 2022*). SARS-CoV-2 delta and omicron variant were obtained from the Research Foundation for Microbial Diseases of Osaka University and the National Institute of Infectious Disease of Japan, respectively.

### Construction of viruses through circular polymerase extension reaction (CPER)

SARS-CoV-2 live-attenuated vaccine candidate strains and B-1 ΔFCS strain have genetic backgrounds similar to that of B-1 strain, in combination with following naturally occurring virulence-attenuating mutations. The mutations in the ORF7a-8, *NSP3*, *NSP14*, and *NSP16* coding regions are the same as those in the genomes of TS virus strains (L50-33, A50-18, and H50-11), described in *Yoshida et al., 2022*. The mutations in the spike FCS ($_{679}$NSPRRARSV$_{687}$ → I) and the NSP1 ($_{32}$GDSVEEVL$_{39}$) are the same as those in the genomes of a laboratory strain described in *Davidson et al., 2020* and of a clinical isolate (accession: LC521925), respectively.

For construction of the strains, we used the CPER method, in which 11 PCR-generated cDNA fragments covering the viral full genome plus another DNA fragment with controlling sequences are stitched into a circular DNA that can produce viral genomic RNA upon introduction to cells (*Okamura et al., 2023*; *Torii et al., 2021*). Primers used to prepare PCR fragments are listed in *Table 4*. The PCR and CPER reaction were performed using PrimeSTAR GXL DNA polymerase (Takara Bio, Cat#

**Table 4.** Primer list.

| Primer ID | Oligonucleotides | Source |
|---|---|---|
| F1_F | CTATATAAGCAGAGCTCGTTTAGTGAACCGTattaaaggtttataccttcccaggtaac | *Torii et al., 2021* |
| F1_R | cagattcaacttgcatggcattgttagtagccttatttaaggctcctgc | *Torii et al., 2021* |
| F2_F | gcaggagccttaaataaggctactaacaatgccatgcaagttgaatctg | *Torii et al., 2021* |
| F2_R | ggtaggattttccactacttcttcagagactggttttagatcttcgcaggc | *Torii et al., 2021* |
| F3_F | gcctgcgaagatctaaaaccagtctctgaagaagtagtggaaaatcctacc | *Torii et al., 2021* |
| F3_R | ggtgcacagcgcagcttcttcaaaagtactaaagg | *Torii et al., 2021* |
| F4_F | caccactaattcaacctattggtgctttggacatatcagcatctatagtagctggtgg | *Torii et al., 2021* |
| F4_R | gtttaaaaacgattgtgcatcagctgactg | *Torii et al., 2021* |
| F5_F | cacagtctgtaccgtctgcggtatgtggaaaggttatggctgtagttgtgatc | *Torii et al., 2021* |
| F5_R | gcggtgtgtacatagcctcataaaactcaggttcccaataccttgaagtg | *Torii et al., 2021* |
| F6_F | cacttcaaggtattgggaacctgagttttatgaggctatgtacacaccgc | *Torii et al., 2021* |
| F6_R | catacaaactgccaccatcacaaccaggcaagttaaggttagatagcactctag | *Torii et al., 2021* |
| F7_F | ctagagtgctatctaaccttaacttgcctggttgtgatggtggcagtttgtatg | *Torii et al., 2021* |
| F7_R | ctagagactagtggcaataaaacaagaaaaacaaacattgttcgtttagttgttaac | *Torii et al., 2021* |
| F8_F | gttaacaactaaacgaacaatgtttgtttttcttgtttttattgccactagtctctag | *Torii et al., 2021* |
| F8_R | gcagcaggatccacaagaacaacagcccttgagacaactacagcaactgg | *Torii et al., 2021* |
| F9_F | ccagttgctgtagttgtctcaagggctgttgttcttgtggatcctgctgc | *Torii et al., 2021* |
| F9_R | caatctccattggttgctcttcatc | *Torii et al., 2021* |
| F10_F | gatgaagagcaaccaatggagattg | *Torii et al., 2021* |
| F10_R | GGAGATGCCATGCCGACCCttttttttttttttttttttttttttgtcattctcctaag | *Torii et al., 2021* |
| Linker_F | cttaggagaatgacaaaaaaaaaaaaaaaaaaaaaaaaaaGGGTCGGCATGGCATCTCC | *Torii et al., 2021* |
| Linker_R | gttacctgggaaggtataaacctttaatACGGTTCACTAAACGAGCTCTGCTTATATAG | *Torii et al., 2021* |
| TS_F6_R | catacaaactgccacTatcacaaccaggcaagttaaggttagatagcactctag | *Yoshida et al., 2022* |
| TS_F7_F | ctagagtgctatctaaccttaacttgcctggttgtgatAgtggcagtttgtatg | *Yoshida et al., 2022* |

R050A). The CPER product was transfected into BHK/hACE2 cells using Lipofectamine LTX (Thermo Fisher Scientific). The cells were incubated at 32°C in a $CO_2$ incubator for 1 week. At this point, the supernatants were collected and transferred to six-well plates which had been pre-seeded with VeroE6/TMPRSS2 cells. The viruses contained in the supernatants of these cells were collected when cytopathic effects (CPEs) were clearly observed.

## Virus titration

The infectious titer of SARS-CoV-2 was determined based on the median tissue culture infectious dose ($TCID_{50}$) or plaque formation assay (PFA). In order to obtain the $TCID_{50}$, virus-containing samples were serially diluted with D-MEM supplemented with 2% FBS. 50 μL of each diluted sample were used to infect Vero cells in 96-well plates. The cells were fixed with 10% formalin after incubating at 32°C for 4 days and stained with crystal violet solution. The $TCID_{50}$ was calculated using the Behrens–Karber method. For the calculation of PFU, 500 μL of the diluted samples were added to confluent Vero cells in six-well plates and incubated at 32°C for 3 hr to allow virus adsorption. The supernatant was removed, and the cells were washed with D-PBS. Subsequently, 2 mL of 1% SeaPlaque agarose dissolved in D-MEM and supplemented with 2% FBS were layered on the cells. The plates were incubated at 32°C in a $CO_2$ incubator for 3 days, fixed with 10% formalin, and stained with crystal violet solution. Visible plaques were counted to calculate PFU.

## Viral proliferation assay at various temperatures

The SARS-CoV-2 B-1 (D614G) strain or vaccine candidate strain was used to infect Vero cells at a multiplicity of infection (MOI) of 0.01. Infected cells were cultured at 32°C or 37°C, and a part of the supernatant was collected daily and stored at –80°C. The viral titers of these samples were determined using the $TCID_{50}$ assay described above. Each experiment was performed in triplicates.

## Quantitative RT-PCR

Viral RNA was extracted from various samples using the IndiSpin Pathogen Kit (INDICAL BIOSCIENCE) following the manufacturer's instructions. RNA was quantified using a DetectAmp SARS-CoV-2 RT-PCR kit (Sysmex). 50 µL of a 100-fold dilution of the extracted RNA were used to perform the reverse transcription and subsequent qPCR reactions in a 7500 Fast Real-time PCR System (Applied Biosystems). The reactions were performed in triplicate.

## Neutralization assay

The neutralizing activity of sera was evaluated using authentic SARS-CoV-2. 100 PFU of authentic virus were mixed with serially diluted serum samples and incubated at 37°C for 1 hr. The mixtures were then transferred to confluent Vero cells in 96-well plates. The infected cells were fixed with 10% formalin and stained with crystal violet solution after an incubation period of 3 days at 32°C. Neutralization titers were calculated as the inverse of the maximum dilution that prevented CPE formation. Additionally, neutralizing antibody titers against SARS-CoV-2 in the serum of monkeys that received three doses of BK2102 were quantified at day 42 with luciferase-expressing pseudovirus carrying the SARS-CoV-2 Wuhan strain spike (Abnova Corporation, Cat# KA6152). Serum samples were fourfold serially diluted and incubated at room temperature for 30 min with Spike-pseudovirus. Subsequently, the mixtures were added into wells containing 293T-ACE2 cells. The luciferase expression of the infected cells was detected with Luciferase Assay System (Promega) after an incubation of 46 hr at 37°C. NT50 neutralization titers were determined by a calibration curve with GraphPad prism.

## ELISA

Half-well protein high-binding 96-well plates (Greiner) were coated with recombinant SARS-CoV-2 (D614G) spike or nucleocapsid proteins (SinoBiologicals) dissolved in PBS (50 ng/100 µL/well) and incubated at 4°C overnight. 1% BSA PBS was used as blocking solution, and 1% BSA PBS-T was used to dilute the sera, antibodies, and streptavidin. Antigen-specific IgG antibodies in hamster sera were detected using horseradish peroxidase (HRP)-labeled goat anti-Syrian hamster IgG H+L Ab (Abcam, Cat# ab6892, 1/30,000 dilution). Biotinylated anti-Syrian hamster IgG1 Ab (Southern Biotech, Cat# 1940-08, 1/100 dilution), IgG2/3 subclass Ab (Southern Biotech, Cat# 1935-08, 1/200,000 dilution), and IgA Ab (Brookwood Biomedical, Cat# 3003a, 1/100 dilution) were used and detected with HRP-labeled streptavidin (Abcam, Cat# ab7403, 1/10,000 dilution). TMB one-component substrate and stopping solutions (Thermodics) were used for the chromogenic reaction, and the optical density (OD) was measured at 450–650 nm with a microplate reader. Endpoint titers of IgG were calculated based on a calibration curve using a five-parameter, nonlinear curve fitting. Samples for which the endpoint titers could be calculated were defined as 'positive', and the mean was calculated using only the titer of positive samples.

## ELISPOT assay

Hamster IFN-γ ELISPOT was performed using the ELISpot Flex: Hamster IFN-γ (ALP) (MABTECH, Cat# 3102-2A) kit according to the manufacturer's instructions. MSIP plates (Millipore) were washed five times with sterile water, coated with Capture mAb (MTH21), and incubated overnight at 4 °C. Coated plates were washed five times with PBS, blocked for 30 min (at room temperature) with RPMI medium supplemented with 10% FBS and antibiotics. 250,000 splenocytes were seeded in each well and stimulated with SARS-CoV-2 spike or nucleocapsid peptide pools, consisting mainly of 15-mer sequences with 11-amino-acid (aa) overlaps (Miltenyi Biotec). Negative controls for non-stimulated cells were included in the test. After stimulation for a day at 37 °C, the spots were detected with the detection mAb (MTH29-biotin) and Streptavidin-ALP. After drying the plate, spots were counted using the ImmunoSpot S5 (Cellular Technology Limited). The average number of spots in the two negative control wells was subtracted from each well stimulated with peptide pools. The result was shown

as the difference in spot-forming cells (SFC)/$10^6$ splenocytes between the negative control and the peptide pool stimulated wells.

## mRNA and LNP production process

A codon-optimized mRNA encoding the SARS-CoV-2 (D614G) S protein was in vitro synthesized and purified following the procedure of Moderna Therapeutics' mRNA-1273 as reference (*Hassett et al., 2019*). The mRNA was encapsulated using a NanoAssemblr Ignite nanoparticle formulation system. The sample was subsequently concentrated to 1 µg/100 µL/dose using Amicon Ultra 4-100K, and filtered through a 0.45 µm membrane for in vivo use.

## Evaluation of immunogenicity in hamsters

The animal experiments were carried out in accordance with the approved protocols corresponding Osaka University's Review Committees (approval no. R02-10) and BIKEN's Review Committees (approval no. ZAE240126-290125-01). Four-week-old male Syrian hamsters were purchased from Japan SLC Inc After a 1-week housing period, they were first anesthetized via inhalation with 3% isoflurane, followed by intraperitoneal anesthesia with a combination of medetomidine, midazolam, and butorphanol (0.3, 4, and 5 mg/kg, respectively). 1000 or $1 \times 10^4$ PFU of BK2102 were administered intranasally in a volume of 20 µL to confine administration to the upper respiratory tract. To evaluate the humoral immune response, blood samples were collected from the facial vein using a lancet (MEDIpoint), and neutralizing antibody titers in the sera were measured as described above. In addition, to evaluate mucosal immunity, nasal wash samples were collected through a plastic catheter using two separate washes of 1 mL of PBS. Only one animal inoculated BK2102 once was not used for IgA measurement because the nasal wash was contaminated with blood. To evaluate the cellular immune response, $1 \times 10^4$ PFU of BK2102-inoculated hamsters were euthanized 1 week post-inoculation, and spleens were collected. The spleens were mechanically crushed with the piston of a syringe and passed through a cell strainer (100 µm). Cell suspensions were treated with RBC lysis buffer (BioLegend) to remove red blood cells, as per the manufacturer's instructions. The spleen cells were then suspended in RPMI medium supplemented with 10% FBS and antibiotics at a concentration of $1 \times 10^6$ cells/mL and stimulated by SARS-CoV-2 spike or nucleocapsid peptide pools, consisting mainly of 15-mer sequences with 11- aa overlaps (Miltenyi Biotec). Negative controls for non-stimulated cells were included in the test. Stimulated cells were incubated at 37°C for 1 day, and supernatants were collected and stored at –80°C until use. IFN-γ and IL-4 were quantified with commercially available ELISA kits (MABTECH AB, Cat# 3102-1H-6 and FineTest, Cat# EHA0001, respectively) following the manufacturer's protocol. Furthermore, IFN-γ-producing cells were identified after a 22 hr stimulation culture with spike or nucleocapsid peptide pools, and analyzed by ELISpot Flex (MABTECH AB). We also performed a challenge assay to investigate whether these immune responses were effective in protecting against other variants. Four weeks post-inoculation with $1 \times 10^3$ PFU or $1 \times 10^4$ PFU of BK2102, $3 \times 10^5$ PFU of SARS-CoV-2 D614G strain, TY41-702 strain (Omicron variant BA.5) or gamma strain were used to challenge the hamsters through the intranasal route to target the lower respiratory tract. The weights of the hamsters were monitored daily, and they were euthanized 4 days after the challenge. Lungs were divided, cut into small pieces, and homogenized with a biomasher II device in 500 µL of D-MEM. Supernatants were collected as lung homogenates after centrifugation at 300 × *g* for 5 min at 4°C. Nasal wash specimens were obtained by flushing 1 mL of D-PBS into the nasal cavity from the trachea in the direction of the nose where it was collected. Infectious virus titers in these samples were evaluated using a PFA, as described above. To evaluate immune response persistence, hamsters inoculated with $1 \times 10^3$ PFU or $1 \times 10^4$ PFU of BK2102 were maintained for 364 days post-infection. Blood samples were collected from the facial vein using a lancet (MEDIpoint), and neutralizing antibody titers were measured. At 420 days after inoculation, a viral challenge assay was performed to evaluate whether the immunity contributing to infection protection was maintained. Hamsters were infected with $3 \times 10^5$ PFU of the SARS-CoV-2 D614G strain, and weight changes were monitored daily for 10 days post-infection.

## Evaluation of immunogenicity in monkeys

The animal experiments were in line with the Institutional Animal Care and Use Committee (IACUC) protocols of Hamri Co, Ltd. (#22-H115). Sixteen male and female cynomolgus monkeys, 2–3 years

old, were purchased from Hamri Co, Ltd. After a 1-week acclimatization housing period, they were sedated with a mixture of ketamine (5 mg/kg) and xylazine (2 mg/kg) administered intramuscularly. Ten million PFU of BK2102 were inoculated intranasally in a volume of 1.6 mL. Four monkeys were inoculated with a single-dose of BK2102, and blood samples were collected at 0, 28, 35, and 42 days after inoculation from the radial vein, femoral vein, or abdominal vena cava using a syringe. Neutralizing antibody titers in the sera were measured as described above to evaluate the humoral immune response. Another 12 monkeys were inoculated with $1 \times 10^7$ PFU of BK2102 or the solvent, receiving three doses given at 2-week intervals. Blood samples were collected 42 days after the last inoculation, and neutralizing antibody titers in the sera were measured.

## Evaluation of BK2102 pathogenicity in hamsters

The animal experiments were carried out in accordance with the approved protocols corresponding Osaka University's Review Committees (approval no. R02-10) and BIKEN's Review Committees (approval no. ZAE240126-290125-01). 4-week-old male Syrian hamsters were obtained from Japan SLC Co. Ltd. After acclimatization, $1 \times 10^4$ PFU of BK2102 or $1 \times 10^4$ PFU of the SARS-CoV-2 D614G strain were used for infection, as described above. These and naïve hamsters were euthanized 3 days post-infection. The heads and lungs were collected and fixed in 10% formalin. Sections of the head were prepared to expose different regions of the nasal cavity and were stained with H&E or an immunohistochemistry (IHC) staining kit (Abcam, Cat# ab64261) using a SARS-CoV-2 spike RBD-specific antibody (Sino Biological, Cat# 40592-T62). The damage score of each section was defined as 0: not remarkable (<10%); 1: minimal (10–50%); and 2: mild (50–70%).

## In vivo passage of BK2102 in hamsters

The animal experiments were carried out in accordance with the approved protocols corresponding Osaka University's Review Committees (approval no. R02-10) and BIKEN's Review Committees (approval no. ZAE240126-290125-01). 4-week-old male Syrian hamsters were obtained from Japan SLC Co. Ltd. After a 1-week acclimatization period, $1 \times 10^5$ PFU of BK2102 or A50-18 strain (a TS mutant isolated in a previous report) in a volume of 20 µL were used to infect the hamsters intranasally. The hamsters were observed daily, and their body weights were measured at 0 and 3 days post-infection. At this point, hamsters were euthanized, and nasal wash specimens were collected with 500 µL D-PBS. After filtration through 0.45 µm and 0.22 µm filters, 100 µL were used to infect a new group of naive hamsters. The passage of inoculum was repeated three times, and nasal wash samples were collected at every passage. The viral genome in these nasal wash samples was quantified via qPCR, as described above. Sanger sequencing was performed to analyze the mutations introduced to generate an attenuated phenotype.

## Generation of human ACE2-transgenic mice

The animal experiments were carried out in accordance with the approved protocols corresponding Osaka University's Review Committees (approval nos. R02-09 and R02-10) and BIKEN's Review Committees (approval no. ZAE240126-290125-01). The transgene was prepared as described previously (*Ikawa et al., 1995*; *Okabe et al., 1997*). Briefly, the hACE2-coding sequence was amplified via PCR with the following primers: 5′-aatctagagccgccgccgccatgtcaagctcttcctggctccttc-3′ and 5′-aaac tcgagctaaaaggaggtctgaacatcatca-3′, using human testis cDNA as the template. The *Xba*I and *Xho*I sites included in the PCR primers were used to introduce the amplified hACE2 cDNA into a pCAG1.1 expression vector (https://www.addgene.org/173685/) containing the chicken *beta-actin* promoter and cytomegalovirus enhancer, the *beta-actin* intron, and the rabbit *globin* poly-adenylation signal. The transgene fragment was excised using *Sac*I and *Pac*I and gel-purified. Transgenic mouse lines were generated by injecting purified transgene fragments into C57BL/6N×C57BL/6N fertilized eggs. A total of 350 DNA-injected eggs were transplanted into pseudopregnant mice, resulting in 32 newborn pups. Three of these were transgenic, and the first line was established as Tg (CAG-hACE2) 01Osb. The Tg mice were kept of a B6D2F1 background. Expression of hACE2 in each organ was confirmed via western blotting. Briefly, organ homogenates were prepared in radioimmunoprecipitation assay buffer containing a proteinase inhibitor cocktail (Thermo Scientific) using a bead mill homogenizer (Fischer Scientific). Protein concentrations were measured via BCA assay (Pierce), and 10 µg of protein

were subjected to SDS-PAGE and subsequent western blotting. hACE2 was detected using a rabbit polyclonal antibody against hACE2 (Abcam, Cat# ab15348) with Can Get Signal solutions (TOYOBO).

## Evaluation of safety in transgenic mice

The animal experiments were carried out in accordance with the approved protocols corresponding Osaka University's Review Committees (approval nos. R02-09 and R02-10) and BIKEN's Review Committees (approval no. ZAE240126-290125-01). 7- to 9-week-old Tg mice were anesthetized via inhalation of 3% isoflurane, then receiving a combination of medetomidine, midazolam, and butorphanol (0.3, 4, and 5 mg/kg, respectively) intraperitoneally. Mice were infected with SARS-CoV-2 in a volume of 20 μL, and their weights were measured daily. Mice that reached humane endpoints, such as difficulty walking or rapid weight loss, were euthanized by bleeding under isoflurane anesthesia. The brain, olfactory bulbs, nasal turbines, and lungs were collected. Mice infected with live-attenuated strains that did not reach the humane endpoint were also euthanized 14 days post-infection.

## Statistical analysis

Two-way ANOVA, one-way ANOVA or Mann–Whitney $U$ test was used to calculate statistical significance. These data were analyzed using GraphPad Prism 9.4.1 software. Statistical significance was set at a p-value$<0.05$.

## Acknowledgements

We appreciate the assistance of Paola Miyazato, Manae Morishima, and Kaori Yamamoto from The Research Foundation for Microbial Diseases of Osaka University (BIKEN). The authors thank Shiho Torii, Chikako Ono, and Yoshiharu Matsuura for their CPER technical support. We would like to thank Mitsuko Mori for generation of human ACE2-transgenic mice. The authors also acknowledge the NGS Core facility of the Genome Information Research Center at the Research Institute for Microbial Diseases of Osaka University for their support with next-generation sequencing analyses. We are grateful to study personnel at BoZo Research Center Inc and Hamri Co, Ltd for their invaluable contributions. This work was conducted as part of 'The Research Foundation for Microbial Diseases of Osaka University Project for Infectious Disease Prevention'. This work was supported by BIKEN, Japan Agency for Medical Research and Development (AMED) grants (JP20pc0101047, JP23fa627002), and Central institute for experimental animals grant (JP23fa627006).

## Additional information

### Competing interests

Mie Suzuki Okutani, Shinya Okamura, Tang Gis, Hitomi Sasaki, Suni Lee, Akiho Kashiwabara, Simon Goto, Mai Matsumoto, Mayuko Yamawaki: employed by BIKEN. Toshiaki Miyazaki, Shiro Takekawa, Hirotaka Ebina: manager of BIKEN. Koichi Yamanishi: director general of BIKEN. The other authors declare that no competing interests exist.

### Funding

| Funder | Grant reference number | Author |
| --- | --- | --- |
| The Research Foundation for Microbial Diseases of Osaka University | | Hirotaka Ebina |
| Japan Agency for Medical Research and Development | JP20pc0101047 | Hirotaka Ebina |
| Japan Agency for Medical Research and Development | JP23fa627002 | Masahito Ikawa |
| Central institute for experimental animals | JP23fa627006 | Masahito Ikawa |

| Funder | Grant reference number | Author |
|--------|------------------------|--------|

The funders had no role in study design, data collection and interpretation, or the decision to submit the work for publication.

## Author contributions

Mie Suzuki Okutani, Data curation, Writing – original draft, Writing – review and editing, conducted and controlled most of the experiments and prepared the manuscript; Shinya Okamura, Data curation, Investigation, Methodology, Writing – original draft, conducted and controlled most of the experiments and prepared the manuscript; Tang Gis, Investigation, supported the experiments related to prolonged protection and in vivo passage assays; Hitomi Sasaki, Investigation, supported the experiments related to prolonged protection and in vivo passage assays; Suni Lee, Investigation, Methodology, conducted the cellular immunity assays; Akiho Kashiwabara, Investigation, constructed various recombinant viruses; Simon Goto, Investigation, Methodology, constructed various recombinant viruses; Mai Matsumoto, Investigation, measured spike-specific IgA titer by ELISA; Mayuko Yamawaki, Investigation, performed neutralizing antibody assay of monkeys using luciferase-expressing pseudovirus carrying the SARS-CoV-2 Wuhan strain's spike protein; Toshiaki Miyazaki, Investigation, Methodology, measured spike-specific IgA titer by ELISA; Tatsuya Nakagawa, Investigation, established Tg-mice; Masahito Ikawa, Methodology, established Tg-mice; Wataru Kamitani, Investigation, Methodology, performed the in vivo passage assays; Shiro Takekawa, Conceptualization, Supervision, Project administration, designed and managed the study; Koichi Yamanishi, Supervision, designed and managed the study; Hirotaka Ebina, Conceptualization, Data curation, Supervision, Funding acquisition, Writing – review and editing, designed and managed the study

## Author ORCIDs

Shinya Okamura ⦿ https://orcid.org/0000-0002-3969-2855
Masahito Ikawa ⦿ https://orcid.org/0000-0001-9859-6217
Hirotaka Ebina ⦿ https://orcid.org/0000-0003-0001-7825

## Ethics

All animal experimental protocols, including anesthesia conditions, endpoints for infection, and euthanasia methods; were reviewed and approved by the corresponding Osaka University's Review Committees (approval no. R02-09 and R02-10), the corresponding BIKEN's Review Committees (approval no. ZAE240126-290125-01), or were in line with the institutional animal care and use committee (IACUC) protocols of Hamri. Co,. Ltd. (#22-H115). All surgeries were performed under triadic anesthesia (medetomidine, midazolam, and butorphanol) or mixed anesthesia (ketamine and xylazine), and every effort was made to minimize suffering.

Reviewer #2 (Public review): https://doi.org/10.7554/eLife.97532.3.sa1
Reviewer #3 (Public review): https://doi.org/10.7554/eLife.97532.3.sa2
Author response https://doi.org/10.7554/eLife.97532.3.sa3

# Additional files

## Supplementary files

MDAR checklist

## Data availability

All data generated or analysed during this study are included in the manuscript and supporting files; source data files have been provided.

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
