## [Editor Report · eLife Assessment]

This is a **valuable** study on the efficacy of a live-attenuated vaccine that was tested in different animal models and the evidence is **convincing**. The study has been strengthened after revisions.

---

## [Referee Report · Reviewer #2 (Public review)]

Summary:

In this manuscript the authors evaluate the attenuation, immunogenicity, and protection efficacy of a live-attenuated SARS-CoV-2 vaccine candidate (BK2102) against SARS-CoV-2.

Strengths:

The authors demonstrate that intranasal inoculation of BK2102 is safe and able to induce humoral and cellular immune responses in hamsters, without apparent signs of damage in the lungs, that protects against homologous SARS-CoV-2 and Omicron BA.5 challenge. Safety of BK2102 was further confirmed in a new hACE2 transgenic mouse model generated by the authors.

Weaknesses:

The authors have addressed my previous comments on the first submission of the document.

---

## [Referee Report · Reviewer #3 (Public review)]

Summary:

Suzuki-Okutani and collogues reported a new live-attenuated SARS-CoV-2 vaccine (BK2102) containing multiple deletion/substitution mutations. They show that the vaccine candidate is highly attenuated and demonstrates great safety profile in multiple animal models (hamsters and Tg-Mice). Of importance, their data show that singe intranasal immunization with BK2102 leads to strong protection of hamsters against D614G and BA.5 challenge in both lungs and URT (nasal wash). Both humoral and cellular responses were induced, and neutralization activity remained for >360 after single inoculation.

Strengths:

The manuscript describes a comprehensive study that evaluates safety, immunogenicity, and efficacy of a new live-attenuated vaccine. Strengths of the study include: (1) strong protection against immune evasive variant BA.5 in both lungs and NW; (2) durability of immunity for >360 days; (3) confirmation of URT protection through a transmission experiment.

While first-generation COVID-19 vaccines have achieved much success, new vaccines that provide mucosal and durable protection remain needed. Thus, the study is significant.

Weaknesses:

Lack of a more detailed discussion of this new vaccine approach in the context of reported live-attenuated SARS-CoV-2 vaccines in terms of its advantages and/or weakness

Antibody endpoint titers could be presented.

Lack of elaboration on immune mechanisms of protection at the upper respiratory tract (URT) against an immune evasive variant in the absence of detectable neutralizing antibodies

Comments on revisions:

In the revised submission, the authors have added new data and have modified the manuscript accordingly. They have reasonably addressed my comments raised in the previous round of review. The quality and clarity of the manuscript are improved.

---

## [Author Response]

The following is the authors’ response to the original reviews.

We sincerely thank the Editor and the Reviewers for their time and effort in thoroughly reviewing our manuscript and providing valuable feedback. We hope we have addressed their comments effectively and improved the clarity of our manuscript as a result.

The major revisions in the updated manuscript are as follows:

(1) Immunization experiments using mRNA in Syrian hamsters were performed (Supplementary figures 2A, B and C).

(2) An ELISPOT assay to evaluate cellular immunity in Syrian hamsters inoculated with BK2102 was conducted (Figure 2F).

(3) IgA titers in BK2102-inoculated Syrian hamsters were successfully measured (Supplementary figure 2B).

(4) New immunogenicity data for BK2102 in monkeys was additionally included (Supplementary figure 3B).

(5) The discussion section has been thoroughly revised to integrate the new data.

These results have been incorporated into the manuscript, and additional text has been added accordingly.

Below, we provide point-by-point responses to the reviewers’ comments and concerns.

**Public Reviews:**

**Reviewer #1:**
(1) A comparative safety assessment of the available m-RNA and live attenuated vaccines will be necessary. The comparison should include details of the doses, neutralizing antibody titers with duration of protection, tissue damage in the various organs, and other risks, including virulence reversal.

We agree with the Reviewer’s comment regarding the lack of data to compare BK2102 with an mRNA vaccine. Unfortunately, we were unable to obtain commercially available mRNA vaccines for research purposes and could not produce mRNA vaccines of equivalent quality. As a result, a direct comparison of the safety profiles of BK2102 and mRNA vaccines was not possible. To address this, we conducted a GLP study with an additional twelve monkeys to evaluate the safety of BK2102. Following three intranasal inoculations of BK2102 at two-week intervals, no toxic effects were observed in any of the parameters assessed, including tissue damage, respiratory rate, functional observational battery (FOB), hematology, or fever. These results are detailed in lines 115-117.

Furthermore, we compared the immunogenicity of BK2102 with that of an in-house prepared mRNA vaccine. The mRNA vaccine was designed to target the spike protein of SARS-CoV-2, and its immunogenicity was evaluated in hamsters. When serum neutralizing antibody titers were found to be comparable between the two, intranasal inoculation of BK2102 induced higher IgA levels in nasal wash samples compared to those from hamsters injected intramuscularly with the self-made mRNA vaccine (Supplementary figures. 2A and B, respectively). Additionally, while the mRNA vaccine induced Th1 and Th2 immune responses, as indicated by the detection of IgG1 and IgG2/3 (Supplementary figure. 2C), BK2102 mainly induced a Th1 response in hamsters. These explanatory sentences have been added to the manuscript (lines 140-150).

(2) The vaccine's effect on primates is doubtful. The study fails to explain why only two of four monkeys developed neutralizing antibodies. Information about the vaccine's testing in monkeys is also missing: What was the level of protection and duration of the persistence of neutralizing antibodies in monkeys? Were the tissue damages and other risks assessed?

We believe that the reason neutralizing antibody titers were observed in only 2 out of 4 monkeys in the immunogenicity study reported in the original manuscript is that only a single-dose was administered. We measured the neutralizing antibody titers in sera collected from monkeys used in the GLP study and confirmed the induction of neutralizing antibody in all 6 monkeys that received three inoculations of BK2102. This data has been included in a new figure (Supplementary figure 3B). While we would have liked to evaluate the persistence of immunity and conduct a protection study in monkeys, limitations related to facility availability and cost prevented us from doing so. As noted in (1), tissue injury and other risk assessments were evaluated in the GLP study, which showed no evidence of tissue injury or other toxic effects. These results are described in lines 113-117.

(3) The vaccine's safety in immunosuppressed individuals or individuals with chronic diseases should be assessed. Authors should make specific comments on this aspect.

In general, live-attenuated vaccines are contraindicated for immunosuppressed individuals or those with chronic conditions, and therefore BK2102 is also not intended for use in these patients.

This information has been added to the Discussion section (lines 309-311).

(4) The candidate vaccine has been tested with a limited number of SARS-CoV-2 strains. Of note, the latest Omicron variants have lesser virulence than many early variants, such as the alfa, beta, and delta strains.

We have added the results of a protection study against the SARS-CoV-2 gamma strain to Supplementary figures 5A and B. No weight loss was observed in BK2102-inoculated hamsters following infection with the gamma strain. These results are described in lines 109-111, 158-162.

(5) Limitations of the study have not been discussed.

We apologize for the ambiguity in the description of the Limitations of this paper. One major limitation of this study is that, despite observing high immunogenicity in hamsters, it remains uncertain whether the same positive results would be achieved in humans. Differences in susceptibility exist between species, which are not solely attributed to weight differences. For instance, while a single dose of 10^3^ PFU of BK2102 was sufficient to induce neutralizing antibodies in hamsters, a higher dose of 10^7^ PFU in monkeys was required to induce antibodies in only about 50% of the monkeys. Additionally, two more challenges in development of BK2102 were added to the discussion. The first was the limited availability of analytical reagents for hamster models, which restricted the detailed immunological characterization of the response. Second, it took time to gather preclinical data due to the space-related restrictions of BSL3 facilities, which delayed the clinical trials for BK2102 until many individuals had already acquired immunity against SARS-CoV-2. It remains to be seen whether our candidate will be optimal for human use, as the immunogenicity of live-attenuated vaccines is generally influenced by pre-existing immunity.

We added these considerations to the discussion section (lines 300-309).

**Reviewer #2:**
No major weaknesses were identified, however, this reviewer notes the following:The authors missed the opportunity to include a mRNA vaccine to demonstrate that the immunity and protection efficacy of their live attenuated vaccine BK2102 is better than a mRNA vaccine.One of the potential advantages of live-attenuated vaccines is their ability to induce mucosalimmunity. It would be great if the authors included experiments to assess the mucosal immunity of their live-attenuated vaccine BK2102.

We agree with the Reviewer’s suggestion regarding the importance of comparing BK2102 with the mRNA vaccine modality and evaluating the mucosal immunity induced by BK2102. In hamsters, under conditions where serum neutralizing antibody titers were equivalent, intranasal inoculation of BK2102 induced higher levels of antigen-specific IgA in nasal wash compared to intramuscular injection of the conventional mRNA vaccine. This new data has been added in Supplementary figures 2A and B, and corresponding sentences have been included in the Results and Discussion sections (lines 140-145, 292-299).

**Reviewer #3:**
Lack of a more detailed discussion of this new vaccine approach in the context of reported live-attenuated SARS-CoV-2 vaccines in terms of its advantages and/or weaknesses.

sCPD9 and CoviLiv, two previously reported live-attenuated vaccines, achieve attenuation through codon deoptimization or a combination of codon deoptimization and FCS deletion. These two strategies affect viral proliferation but do not directly impact virulence. In contrast, the temperature sensitivity-related substitutions in NSP14 included in BK2102 selectively restrict the infection site, reducing the likelihood of lung infection and providing a safety advantage over the other live-attenuated vaccines. As mentioned in the response to comment (5) of Reviewer #1, a limitation of BK2102 is that its development began later than that of the previously reported live-attenuated vaccines. Consequently, we must consider the impact of pre-existing immunity in future human trials. Based on these points, we have added sentences discussing the advantages and disadvantages to the Discussion section (lines 302-305, 312-319).

Antibody endpoint titers could be presented.

Thank you for your suggestion. We calculated the antibody endpoint titers for Figure 2A and included the results in lines 105-107 of the revised manuscript.

Lack of elaboration on immune mechanisms of protection at the upper respiratory tract (URT) against an immune evasive variant in the absence of detectable neutralizing antibodies.

We appreciate the comment. The potential role of cellular and mucosal immunity in protection has been discussed in more detail in the revised manuscript, specifically in lines 283-295. According to the reference we initially cited, Hasanpourghadi et al. evaluated their adenovirus vector vaccine candidates and reported that the protection was enhanced by co-expression of the nucleocapsid protein rather than relying solely on the spike protein (Hasanpourghadi et al., Microbes Infect, 2023). Therefore, cellular immunity against the nucleocapsid and/or other viral proteins induced by BK2102 may also contribute to protection, as evidenced by more pronounced cellular immunity to the nucleocapsid detected through ELISPOT assay. Moreover, antigen-specific mucosal immunity was successfully detected in additional studies. The involvement of mucosal immunity in protection against mutant strains has been documented in the previously cited reference (Thwaites et al., Nat Commun, 2023). We have included these new data in Figure 2F and Supplementary figure 2B. Additionally, the results and discussion regarding the mechanisms of protection in the upper respiratory tract, in the absence of detectable neutralizing antibodies, have been incorporated into the revised lines 136-139, 143-145 and 283-295, respectively.

**Recommendations for the authors:**

**Reviewer #2:**
Figure 1: Please include the LOD and statistical analysis in both panels. Please consider passaging the virus in Vero cell s, approved for human vaccine production, to assess the stability of BK2102 after serial passage in vitro, which is important for its implementation as a live-attenuated vaccine. The authors should consider evaluating viral replication in different cell lines, and also assessing the plaque phenotype.

Thank you for your valuable comments. First, we have added the statistical analysis and the limit of detection (LOD) to Figure 1. In response to the comments regarding the stability of BK2102 after serial passage in Vero cells, as well as its replication and plaque phenotype in different cell lines, we manufactured test substances for GLP studies and clinical trials by passaging BK2102 in Vero cells, which are approved for human vaccine production. We confirmed that BK2102 is stable (data not shown). Additionally, we verified that BK2102 replicates in BHK, Vero E6, and Vero E6/TMPRSS2 cells, in addition to Vero cells. Among these options, we selected Vero cells due to their high proliferative capacity and ability to produce clear plaques.

Figure 2: Please, include statistical analysis in panels A, B, and D. Please, include the LOD in panels A and D. Please, include viral titers from these experiments in hamsters and NHPs.

First, we would like to note that Figure 2D has been replaced by Figure 2C in the revised manuscript, and the data on neutralizing antibody titers in non-human primates (NHPs), originally presented as Figure 2C, have been moved to the Supplementary figure 3A.

We have added the statistical analysis to Figure 2B and C, as well as the LOD to Figure 2C. Figure 2A (Spike-specific IgG ELISA) was intended for qualitative evaluation based on OD values, so the LOD was not defined. We have also added a detailed description of virus titer in the Methods section under the headings “Evaluation of Immunogenicity in Hamsters” and “Evaluation of Immunogenicity in Monkeys”, and updated the information in the Figure legends of the revised manuscript (lines 451, 459, 468-474, 566-567, 576-578, 582-584, 661-662).

Figure 3: Please, include the viral titers of the challenge virus in the NT and lungs.

We have added the virus titers for the challenge experiments to the Results section under the heading “BK2102 induced protective immunity against SARS-CoV-2 infection” (lines 168-174).

Figure 4: Please, include statistical analysis in panels B and C and evaluate viral titers.

We have added the statistical analysis to Figure 4B and C. Unfortunately, all samples in Figure 4 were fixed in formalin for histopathological examination, so virus titers could not be measured. However, in past experiments, we measured viral titers in the nasal wash samples and lungs of hamsters three days post-infection with D614G and BK2102. We confirmed that infectious virus was detected in both the nasal wash and lungs of the hamsters infected with D614G strain (2.9 log10 PFU/mL and 5.3 log10 PFU/g, respectively), but not in the lungs of the hamsters with BK2102. The viral titers in the nasal wash of BK2102-infected hamsters were equivalent to those of the hamsters infected with the D614G wild-type strain (3.0 log10 PFU/mL). However, we did not include this data to the revised manuscript.

Figure 5: Please, include viral titers in different tissues with the different vaccines (panels A and B). Please, include the body weight changes. Finally, please, consider the possibility of challenging the vaccinated mice with the same SARS-CoV-2 strains used in the manuscript to demonstrate similar protection efficacy in this new ACE2 transgenic mice.

The different tissues of Tg mice were not sampled, as no gross abnormalities were observed in organs other than lungs and brains during necropsy. We have added new data on the body weight of Tg mice after infection to Supplementary figures 9B and 9C in the revised manuscript, along with additional lines in the Results section (lines 228-230 and 247-248). Although we do not know the reason, we have observed that immunization of this animal model does not lead to an increase in antibody titers. Therefore, we do not consider this animal model suitable for the protection study as you suggested. However, it could be useful in passive immunization experiments.

Supplementary Figure 1: Since most of the manuscript focuses on BK2102, the authors should consider removing the other live-attenuated vaccines (Supplementary Figure 1A).

We agree with the Reviewer’s suggestion and have simplified the description for Supplementary Figure 1A (lines 93-97).

Supplementary Figure 3: Please, include statistical analysis.

In the revised manuscript, Supplementary Figure 3 from the original manuscript has been moved to Supplementary Figure 2D. The IgG subclass ELISA was intended for a qualitative evaluation based on OD values, and therefore the results were included in the Supplementary figure. However, we realized the description was not clear, so we added further clarification in the Results section (lines 145-147).

Supplementary Figure 4: Please, include the viral titers in both infected and contact hamsters from this experiment.

In the revised manuscript, Supplementary Figure 4 in the original manuscript has been moved to Supplementary Figure 6. Unfortunately, due to limited breeding space for the hamsters, we were unable to prepare groups for the evaluation of viral titer, and instead prioritized evaluation by body weight.

**Reviewer #3:**
(1) It would be helpful to discuss this new vaccine in the context of other reported live-attenuated vaccines in terms of its advantages and/or disadvantages.

Please refer to our response to the Reviewer’s “first comment” above, as well as to the response in Public comment (5) of Reviewer #1. The modifications made in the manuscript are described in lines 302-305 and 312-319.

(2) Figure 2A: end-point titers could be presented, other than OD values.

This comment is addressed in the reviewer’s second public comment. The endpoint titer has been included in lines 105-107 of the revised manuscript.

(3) Figure 2C: it is unclear why only 2 out of 4 NHPs show neutralization titers. This could be moved to a supplementary figure.

As suggested by the Reviewer, Figure 2C of the original manuscript has been moved to Supplementary Figure 3A in the revised manuscript. In response Public comment (2) from Reviewer #1, we have also added new data on neutralizing antibodies in the monkeys as Supplementary figure 3B.

(4) Figures 2E-F: bulk measurement of cytokine production in supernatants is not an optimal way to measure vaccine-induced Ag-specific T cells. ELISPOT or ICS are better. T-cell ELSIPOT for hamsters is available. This should at least be discussed.

Please refer to our response to this Reviewer’s third public comment. We have added the new results in Figure 2F of the revised manuscript.

(5) It is quite interesting that no N-specific cellular response was observed, given that it is a live-attenuated vaccine. What about N-specific binding Abs?

We conducted the ELISPOT assay as suggested by the Reviewer and detected cellular immunity against both spike and nucleocapsid proteins (Figure 2F). We did not examine nucleocapsid-specific antibodies, as they do not contribute to the neutralizing activity; however, nucleocapsid-specific cellular immunity was confirmed.

(6) Figure 3: limit of detection for virological assays could be labeled.

We have added the LOD in Figures 3C, D, F and G.

(7) Figures 3E-F: it is interesting to see that the vaccine elicits almost complete protection at URT against BA.5, despite no BA.5 neutralizing titers being detected at all. What mechanism of URT protection by BK2102 would the authors speculate? T cells or other Ab effector functions?

Please refer to the response to this Reviewer’s third public comment. We have added new results regarding cellular and mucosal immunity (Figure 2F and Supplementary figure 2B) and discussed the mechanisms of protection in the upper respiratory tract in the absence of detectable neutralizing antibodies (lines 136-139, 143-145 and 283-295, respectively).

(8) Figure 3I: the durability of protection is a strength of the study. Other than body weight changes, what about viral loads in the animals after the challenge?

We primarily assessed the effect of the vaccine by monitoring changes in body weight, as the differences compared to the naïve group were clear. Unfortunately, we did not collect samples at different time points throughout the study, which prevented us from evaluating the viral titers.

In addition, we made corrections to several other sections identified during the revision process. The revised parts are as follows:

- In the Methods section under the title “Evaluation of BK2102 pathogenicity in hamsters”, the infectious virus titer of D614G strain has been corrected (line 478).

- In the Methods section under the title “In vivo passage of BK2102 in hamsters”, infectious virus titer of BK2102 and A50-18 strain has been corrected (line 487).

- The collection time of splenocytes after inoculation has been corrected in the figure legend of Figure 2D, (line 583).

- There was an error in Figure 2D. The figure has been replaced with the appropriate version.

- A new reference on NSP1 deletion (Ueno et al., Virology, 2024) has been added to the references.

- Several methods have been described more clearly.